# UniMoMo: Unified Generative Modeling of 3D Molecules for *De Novo* Binder Design

**Xiangzhe Kong** [1 2]   **Zishen Zhang** [1]   **Ziting Zhang** [3]   **Rui Jiao** [1 2]
**Jianzhu Ma** [4 2]   **Wenbing Huang** [5 6]   **Kai Liu** [7]   **Yang Liu** [1 2]

## Abstract

The design of target-specific molecules such as small molecules, peptides, and antibodies is vital for biological research and drug discovery. Existing generative methods are restricted to single-domain molecules, failing to address versatile therapeutic needs or utilize cross-domain transferability to enhance model performance. In this paper, we introduce **Uni**fied generative **Mo**deling of 3D **Mo**lecules (UniMoMo), the first framework capable of designing binders of multiple molecular domains using a single model. In particular, UniMoMo unifies the representations of different molecules as graphs of blocks, where each block corresponds to either a standard amino acid or a molecular fragment. Subsequently, UniMoMo utilizes a geometric latent diffusion model for 3D molecular generation, featuring an iterative full-atom autoencoder to compress blocks into latent space points, followed by an E(3)-equivariant diffusion process. Extensive benchmarks across peptides, antibodies, and small molecules demonstrate the superiority of our unified framework over existing domain-specific models, highlighting the benefits of multi-domain training.

## 1. Introduction

Designing molecules that bind to target proteins is a fundamental task in biological research and drug discovery, as

[1]Dept. of Comp. Sci. & Tech., Tsinghua University [2]Institute for AI Industry Research (AIR), Tsinghua University [3]Dept. of Automation, Tsinghua University [4]Dept. of Electronic Engineering, Tsinghua University [5]Gaoling School of Artificial Intelligence, Renmin University of China [6]Beijing Key Laboratory of Research on Large Models and Intelligent Governance [7]Bytedance Project Voyager Team. Correspondence to: Wenbing Huang <hwenbing@126.com>, Kai Liu <liukai0824@bytedance.com>, Yang Liu <liuyang2011@tsinghua.edu.cn>.

*Proceedings of the 42^nd International Conference on Machine Learning*, Vancouver, Canada. PMLR 267, 2025. Copyright 2025 by the author(s).

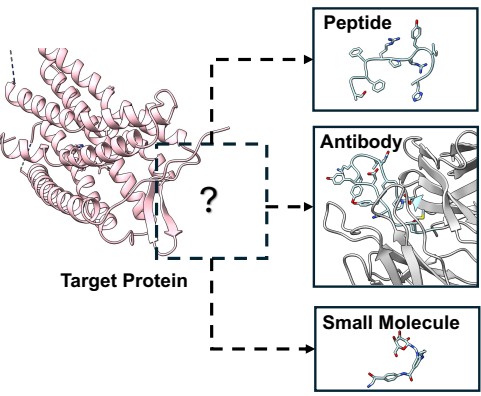

*Figure 1.* Given the binding site on the target protein, our UniMoMo is capable of generating diverse molecular binders, including peptides, antibodies, and small molecules.

the resulting bindings can induce or block specific biological pathways of interest (Glögl et al., 2024; Wang et al., 2021). Various types of molecules have been exploited for different purposes because of their unique biochemical properties. For instance, small molecules are ideal for oral administration due to their high absorption efficiency (Beck et al., 2022), peptides excel in intracellular targeting thanks to their cell-penetrating capability (Derakhshankhah & Jafari, 2018), and antibodies are favored for treating severe diseases like cancer because of their high specificity (Basu et al., 2019). The distinct roles of these molecular types have led to domain-specific generative models. In particular, for small molecules, research has focused on autoregressive models (Peng et al., 2022), diffusion-based approaches (Guan et al., 2023a; Schneuing et al., 2024), Bayesian flow networks (Qu et al., 2024), and voxel-based representations (Pinheiro et al., 2024). For peptides and antibodies, diffusion models (Luo et al., 2022; Kong et al., 2024b), iterative refinement (Jin et al., 2021; Kong et al., 2023a;b) and multi-modal flow matching (Li et al., 2024b; Lin et al., 2024c) are prevalent.

We argue that a unified generative framework for all types of molecules offers distinct advantages over the current domain-specific paradigm. From an application standpoint, such a framework enables the exploration of multiple drugs spanning diverse molecular types for a single target, addressing varied therapeutic needs. From a machine learning

perspective, unified modeling leverages larger and more diverse datasets, better exploiting available data. Moreover, this methodology is well founded, as the principles underlying molecular binder design are consistent across different molecular types, including interactions with the target protein (Bissantz et al., 2010) and adherence to geometric constraints like bond lengths, angles, and clashes (Lide, 2004), all governed by the same physiochemical rules.

However, designing a unified generative framework poses significant challenges due to the distinct representation of different molecular types. Small molecules comprise various functional groups assembled in diverse patterns (Qu et al., 2024), while peptides and antibodies are composed of amino acids arranged linearly (Kong et al., 2024b), with antibodies further featuring distinct regions defined by their functional roles (Kong et al., 2023a). A straightforward yet suboptimal approach is to treat all molecules as atomic graphs regardless of distinct types. However, this ignores the intrinsic hierarchical priors of molecular structures (e.g., benzene rings or standard amino acids) and results in high computational complexities for larger systems like peptides and antibodies. Conversely, relying solely on a vocabulary of common substructures, without preserving atomic details, lacks transferability, as the shared principles of molecular binder design rely heavily on full-atom geometry. Thus, developing an effective and efficient unified generative framework that incorporates both atomic details and hierarchical priors remains an open and pressing challenge.

In this paper, we propose **Uni**fied generative **Mo**deling of 3D **Mo**lecules (UniMoMo), the first framework capable of designing binders across multiple molecular types using a single model. Our approach begins by representing 3D molecules as graphs of blocks, where each block corresponds to either a standard amino acid or a molecular fragment (Figure 2). Fragments, extracted from small molecules and non-standard amino acids, are identified using the principal subgraph algorithm (Kong et al., 2022). This representation preserves both the full-atom geometry and the hierarchical structure of building blocks, which are critical for modeling interactions and local geometries (Figure 1). Built upon the unified representations, we leverage a geometric latent diffusion model for efficient and effective generative modeling. First, an iterative full-atom autoencoder is trained to compress each block into a latent representation consisting of a low-dimensional hidden state and a spatial coordinate, then reconstruct the full-atom geometries from the latent point cloud with two-stage decoding. Subsequently, a diffusion model operates in the latent space to generate the E(3)-invariant latent states and the E(3)-equivariant coordinates.

The architecture design of our model offers the following two key advantages. First, the compressed latent space

allows the diffusion model to bypass the intra-block dependencies within building blocks, focusing instead on global arrangements, with the fine-grained reconstruction of local full-atom details handled by the decoder module of the autoencoder. Second, by performing the diffusion process in the latent space, with reduced dimensionality and graph size, the model achieves significant improvements in training and inference efficiency.

We conduct extensive experiments on benchmarks of peptides, antibodies, and small molecules. The results demonstrate the superiority of our unified framework over the domain-specific models, and also illustrate the benefits of training a unified model with multi-domain molecular data. Furthermore, we showcase the ability of UniMoMo to generate binders targeting the same protein pocket across different molecular types with desired affinity.

## 2. Related Work

**Small Molecule Design** Structure-based drug design (SBDD) has advanced with deep generative models. Early voxel-based methods, like LIGAN (Masuda et al., 2020) and 3DSBDD (Luo et al., 2021), predict atom densities and types using VAEs or autoregressive approaches. Equivariant Networks (Satorras et al., 2021) enable direct 3D atomic position generation, as in Pocket2Mol(Peng et al., 2022) and GraphBP (Liu et al., 2022). Diffusion models (Ho et al., 2020) further improved SBDD with methods like TargetDiff (Guan et al., 2023a), DiffBP (Lin et al., 2025), and DiffSBDD (Schneuing et al., 2024). Recent works, including FLAG (Zhang et al., 2023), D3FG (Lin et al., 2024a), and DecompDiff (Guan et al., 2023b), leverage domain knowledge, while MolCraft (Qu et al., 2024) and VoxBind (Pinheiro et al., 2024) introduce Bayesian Flow Networks (Graves et al., 2023) and walk-jump sampling (Frey et al., 2024). Our work relates closely to diffusion and fragment-based models, as we employ principal subgraphs (Kong et al., 2022) to decompose molecules into blocks within our unified generative framework.

**Peptide Design** Peptide design has shifted from energy-based sampling (Bhardwaj et al., 2016; Hosseinzadeh et al., 2021) to deep generative models (Li et al., 2024a). PepFlow (Li et al., 2024b) and PPFlow (Lin et al., 2024c) similarly apply multi-modal flow matching over residue types, orientations, $C_\alpha$ positions, and side-chain dihedral angles. PepGLAD (Kong et al., 2024b) employs geometric latent diffusion models to co-design the sequence and full-atom structure of peptides. Additional efforts have explored specific peptide subtypes, such as $\alpha$-helical peptides (Xie et al., 2024) and D-peptides (Wu et al., 2024). Our work is closely related to PepGLAD, leveraging geometric latent diffusion, but with the novel inclusion of iterative design and bond prediction within the full-atom autoencoder.

**Antibody Design** Antibody design, which mainly focuses on the Complementarity-Determining Regions (CDRs), has also evolved from forcefield-based sampling (Adolf-Bryfogle et al., 2018) to deep learning (Saka et al., 2021; Akbar et al., 2022; Martinkus et al., 2024; Jin et al., 2021). MEAN (Kong et al., 2023a) and DiffAb (Luo et al., 2022) generate antigen-specific antibodies using iterative decoding or diffusion, while dyMEAN (Kong et al., 2023b) handles full-atom generation in scenarios of unknown antibody docking poses. HERN (Jin et al., 2022) employs hierarchical graphs and auto-regressive decoding, while GeoAB (Lin et al., 2024b) incorporates physical constraints. Other approaches explore pretraining (Wu & Li, 2024; Gao et al., 2023; Zheng et al., 2023), and explicit optimization on energies to enhance performance (Zhou et al., 2024).

**Unified Modeling** GET (Kong et al., 2024a) pioneers the unified modeling of biomolecules for molecular interaction prediction, showcasing zero-shot generalizability. Subsequent works further explore unified language models (Zheng et al., 2024) and inverse folding models (Gao et al., 2024). Recently, Alphafold 3 (Abramson et al., 2024) achieves groundbreaking accuracy in structure prediction across diverse biomolecular types. Our work is inspired by GET and Alphafold 3, which highlight the transferability of cross-molecular interactions, but focuses on the more challenging task of *de novo* molecular binder design.

# 3. Method: UniMoMo

In the following section, we begin by introducing the necessary notations and our unified molecular representation in §3.1. Next, we elaborate on the two core components of our UniMoMo: 1) A full-atom variational autoencoder that compresses building blocks into latent points and iteratively reconstructs atom-level details (§3.2); 2) A geometric diffusion model that operates within the latent space (§3.3). The overall framework is illustrated in Figure 2.

## 3.1. Notations and Unified Representation

We represent a molecule as a graph of blocks $\mathcal{G} = (\mathcal{V}, \mathcal{E})$. $\mathcal{V} = \{(\boldsymbol{A}_i, \vec{\boldsymbol{X}}_i) \mid \boldsymbol{A}_i \in \mathbb{Z}^{n_i}, \vec{\boldsymbol{X}}_i \in \mathbb{R}^{n_i \times 3}\}$, where each block $i$ has $n_i$ atoms, with $\boldsymbol{A}_i$ and $\vec{\boldsymbol{X}}_i$ representing their element types and coordinates, respectively. $\mathcal{E} = \{(\boldsymbol{B}_i, \boldsymbol{B}_{ij}) \mid i \neq j\}$, where $\boldsymbol{B}_i$ and $\boldsymbol{B}_{ij}$ denote intra-block and inter-block chemical bonds, respectively. For simplicity, we use matrix notation here, but the atoms within each block are unordered. To decompose molecules into blocks, we first identify all standard amino acids as individual blocks. In other cases, such as non-standard amino acids and small molecules, we leverage the principal subgraph algorithm (Kong et al., 2022) for decomposition, as illustrated in Figure 2. This algorithm begins with an atom-level graph

and iteratively merges the frequent neighboring nodes into common building blocks such as benzene and indole (Appendix A). The resulting vocabulary of blocks, denoted by $\mathbb{V}$, comprises standard amino acids and the extracted principal subgraphs. Consequently, each block $i$ in the molecular graph can be assigned with a type $s_i \in \mathbb{V}$. For better controllability, we introduce a binary prompt for each block, with 1 indicating the generation of natural amino acids (AAs), and 0 indicating either AAs or molecular fragments. During benchmarking for peptides and antibodies, all blocks are assigned a prompt of 1 to enforce AA-only generation. For small molecules, we set the prompt to 0, allowing flexible sampling from a broader set of molecular fragments.

The molecular binder and the binding site of the target protein are denoted as two graphs of blocks, $\mathcal{G}_x$ and $\mathcal{G}_y$, respectively. Any optional context, such as the framework regions of antibodies, are also included in $\mathcal{G}_y$. In this paper, we address *de novo* binder design, which involves generative modeling of $p(\mathcal{G}_x \mid \mathcal{G}_y)$, namely the conditional distribution of the molecular binder given the context environment. To avoid information leakage of reference binders, the binding site is defined as residues within 10Å distance to the binder based on $C_\beta$ atoms, following previous work (Peng et al., 2022; Kong et al., 2024b), or based on the center of mass for those blocks without $C_\beta$ atoms.

## 3.2. Iterative Full-Atom Variational AutoEncoder

We first propose an iterative full-atom variational autoencoder (VAE), comprising an encoder $\mathcal{E}_\phi$ and a decoder $\mathcal{D}_\xi$ (Vincent et al., 2010), to establish a mapping between the full-atom geometries and a compact latent space.

**Encoder** The encoder $\mathcal{E}_\phi$ projects the binder $\mathcal{G}_x$ and the binding site $\mathcal{G}_y$ into latent point clouds $\mathcal{Z}_x$ and $\mathcal{Z}_y$:

$$\mathcal{Z}_x = \mathcal{E}_\phi(\mathcal{G}_x), \qquad \mathcal{Z}_y = \mathcal{E}_\phi(\mathcal{G}_y), \tag{1}$$

where $\mathcal{Z}_x = \{(\boldsymbol{z}_i, \vec{\boldsymbol{z}}_i)\}$ contains the latent states $\boldsymbol{z}_i \in \mathbb{R}^d$ ($d = 8$ in this paper) and coordinates $\vec{\boldsymbol{z}}_i \in \mathbb{R}^3$ of block $i$ in $\mathcal{G}_x$. Both $\boldsymbol{z}_i$ and $\vec{\boldsymbol{z}}_i$ are sampled from the encoded distribution $\mathcal{N}(\boldsymbol{z}_i; \boldsymbol{\mu}_i, \boldsymbol{\sigma}_i)$ and $\mathcal{N}(\vec{\boldsymbol{z}}_i; \vec{\boldsymbol{\mu}}_i, \vec{\boldsymbol{\sigma}}_i)$ using the reparameterization trick (Kingma, 2013). The latent representation $\mathcal{Z}_y$ for the binding site follows a similar definition. Note that the binder and the binding site are encoded independently, without access to each other's information. Such implementations align with test-time applications where only the binding site $\mathcal{G}_y$, without a binder $\mathcal{G}_x$, is input as the condition. To regularize the scale of the latent states $\boldsymbol{z}_i$ and coordinates $\vec{\boldsymbol{z}}_i$, we implement Kullback–Leibler (KL) divergence constraints relative to the prior distributions $\mathbb{N}(\boldsymbol{0}, \boldsymbol{I})$ and $\mathbb{N}(\vec{\boldsymbol{r}}_i, \boldsymbol{I})$ (Kong et al., 2024b), where $\vec{\boldsymbol{r}}_i$ represents the center of mass for all atoms in block $i$:

$$\begin{aligned}\mathcal{L}_{KL}(i) = &\; \lambda_1 \cdot D_{\mathrm{KL}}(\mathcal{N}(\boldsymbol{0}, \boldsymbol{I}) \| \mathcal{N}(\boldsymbol{\mu}_i, \mathrm{diag}(\boldsymbol{\sigma}_i))) \\ &+ \lambda_2 \cdot D_{\mathrm{KL}}(\mathcal{N}(\vec{\boldsymbol{r}}_i, \boldsymbol{I}) \| \mathcal{N}(\vec{\boldsymbol{\mu}}_i, \mathrm{diag}(\vec{\boldsymbol{\sigma}}_i))),\end{aligned} \tag{2}$$

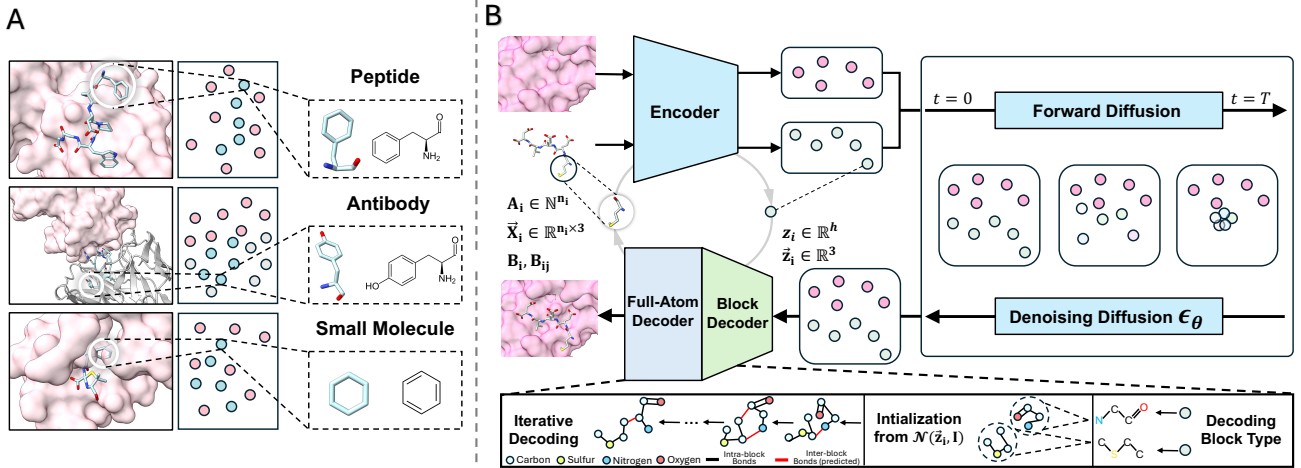

*Figure 2.* **Overview of our UniMoMo. (A)** Graph of blocks as the unified representation for peptides, antibodies, and small molecules. **(B)** The proposed unified generative framework for 3D molecular binder design involves an iterative full-atom autoencoder and a diffusion model implemented in the latent space. The autoencoder compresses the atomic details of each block into single latent points and reconstructs them by first predicting block types for the latent points, followed by iterative generation of the full-atom geometries.

where $D_{\mathrm{KL}}$ denotes the KL divergence, $\lambda_1$ and $\lambda_2$ control the strength of regularization for the latent states and coordinates, respectively. During training, we further perturb $\vec{z}_i$ by adding random noise sampled from $\mathcal{N}(\mathbf{0}, \boldsymbol{I})$ before feeding it into the decoder. This is crucial because the diffusion module, operating in the latent space, is hard to generate precise locations, which introduces errors in the latent coordinates during the generation process. By incorporating noise during training in advance, the decoder learns to handle these imperfections effectively with much higher robustness.

**Decoder** The decoder $\mathcal{D}_\xi$ reconstructs the full-atom geometry of the binder in two stages: decoding block types and reconstructing atomic coordinates. First, the decoder predicts the block types $\{s_i\}$ for the latent points of the binder using both $\mathcal{Z}_x$ and $\mathcal{Z}_y$, yielding the atom types and chemical bonds within each block via a vocabulary lookup:

$$\{s_i\} = \mathcal{D}_{\xi_1}(\mathcal{Z}_x, \mathcal{Z}_y), \qquad \boldsymbol{A}_i, \boldsymbol{B}_i = \mathrm{Lookup}(s_i, \mathbb{V}), \quad (3)$$

where $s_i$, $\boldsymbol{A}_i$, and $\boldsymbol{B}_i$ denote the decoded block type, the atom types, and the chemical bonds of block $i$, respectively. Next, the atomic coordinates are reconstructed in an iterative manner by a structure module, similar to a light-weight flow matching process (Lipman et al., 2023):

$$\{(\boldsymbol{H}_i^t, \vec{\boldsymbol{V}}_i^t)\} = \mathcal{D}_{\xi_2}(\{(\boldsymbol{A}_i, \boldsymbol{B}_i, \vec{\boldsymbol{X}}_i^t)\}, \mathcal{Z}_x, \mathcal{Z}_y, \mathcal{G}_y, t), \quad (4)$$

$$\vec{\boldsymbol{X}}_i^{t-\Delta t} = \vec{\boldsymbol{X}}_i^t + \Delta t \vec{\boldsymbol{V}}_i^t, \quad (5)$$

where $\boldsymbol{H}_i^t$ and $\vec{\boldsymbol{V}}_i^t$ are the atomic hidden states and the vector fields for each atom in block $i$. The time variable $t$ decreases from 1.0 to 0.0 with step size $\Delta t = 0.1$ in this paper, totaling 10 iterations. At $t = 1.0$, the initial coordinates $\vec{\boldsymbol{X}}_i^1$ are sampled from a Gaussian distribution $\mathcal{N}(\vec{z}_i, \boldsymbol{I})$ for each block $i$. The chemical bonds between

atoms in different blocks are also predicted based on the spatial neighborhood (distance below 3.5Å) of atoms:

$$b_{ij,pq}^t = \mathrm{Softmax}(\mathrm{MLP}(\boldsymbol{h}_{i,p}^t + \boldsymbol{h}_{j,q}^t)), i \neq j, \quad (6)$$

where $b_{ij,pq}^t$ represents the type of chemical bond (e.g., single, double, triple, or none) between the $p$-th atom in block $i$ and the $q$-th atom in block $j$. Furthermore, $\boldsymbol{h}_{i,p}^t = \boldsymbol{H}_i^t[p]$ and $\vec{\boldsymbol{x}}_{i,p}^t = \vec{\boldsymbol{X}}_i^t[p]$ denotes the hidden states and coordinates of the $p$-th atom in block $i$, respectively. MLP denotes a 3-layer multi-layer perceptron with SiLU activation (Hendrycks & Gimpel, 2016), the input of which is the summation of $\boldsymbol{h}_{i,p}^t$ and $\boldsymbol{h}_{j,q}^t$ to enable commutativity. Bond prediction is restricted to spatially neighboring atoms within 3.5Å, determined by the distance $d_{ij,pq}^t = \|\vec{\boldsymbol{x}}_{i,p}^t - \vec{\boldsymbol{x}}_{j,q}^t\|_2$. While potential inconsistency between predicted bonds and distances exists, they could be easily distinguished and discarded (Appendix G). The reconstruction loss includes cross entropy (CE) for block types and bond types, as well as the mean square error (MSE) for vector fields:

$$\begin{aligned} \mathcal{L}_{\mathrm{rec}}(i) = {} & \mathrm{CE}(p(\hat{s}_i), p(s_i)) \\ & + \mathbb{E}_{t \sim U(0,1)}\left[\sum_{j,p,q} \mathrm{CE}(p(\hat{b}_{ij,pq}^t), p(b_{ij,pq}^t))\right] \\ & + \mathbb{E}_{t \sim U(0,1)}\left[\mathrm{MSE}(\hat{\vec{\boldsymbol{V}}}_i^t, \vec{\boldsymbol{V}}_i^t)\right], \end{aligned} \quad (7)$$

where $p(\hat{s}_i), p(\hat{b}_{ij,pq}^t)$, and $\hat{\vec{\boldsymbol{V}}}_{i,\mathrm{gt}}^t$ denote the ground-truth block types, bond types, and vector fields, respectively. To encourage better reconstruction of the interactions and local geometries, we also implement a pair-wise distance loss $\mathcal{L}_{\mathrm{dist}}$ on the neighborhood for each atom. As the geometries are still in chaos on large time values, we only implement

this loss when $t < 0.25$:

$$\mathcal{L}_{\text{dist}}(i, t) = \frac{\sum_{p \in \mathcal{A}_i, (j,q) \in \mathcal{N}_i(p)} |\hat{d}^0_{ij,pq} - d^0_{ij,pq}|}{\sum_{p \in \mathcal{A}_i} |\mathcal{N}_i(p)|}, \quad (8)$$

$$\mathcal{L}_{\text{dist}}(i) = \mathbb{E}_{t \sim U(0,1)}[\mathbb{I}_{t<0.25} \cdot \mathcal{L}_{\text{dist}}(i, t)] \quad (9)$$

where $\mathcal{N}_i(p) = \{(j, q) \mid \hat{d}^0_{ij,pq} < 6.0\}$ denotes all atoms within 6Å distance to the $p$-th atom in block $i$ in the ground-truth structure. To make the latent point cloud of the binding site $\mathcal{G}_y$ more informative, we also implement the KL divergence and reconstruction loss on 5% of the binding site residues. Denoting these randomly selected residues as $\tilde{\mathcal{G}}_y$, we have the overall objective for the autoencoder as follows:

$$\mathcal{L}_{\text{AE}} = \sum_{i \in \mathcal{G}_x \cup \tilde{\mathcal{G}}_y} \frac{\mathcal{L}_{\text{KL}}(i) + \mathcal{L}_{\text{rec}}(i) + \lambda_{\text{dist}} \mathcal{L}_{\text{dist}}(i)}{|\mathcal{G}_x \cup \tilde{\mathcal{G}}_y|}, \quad (10)$$

where $\lambda_{\text{dist}}$ balances the contribution of the distance loss.

Both the encoder and the decoder are parameterized by an equivariant transformer (Jiao et al., 2024), with details in Appendix F. Training employs teacher forcing, where the structure module receives ground-truth atom types $\hat{A}_i$ and intra-block bonds $\hat{B}i$, and, with 50% probability, inter-block bonds $\hat{B}ij$. During inference, full-atom structures and inter-block bonds are generated simultaneously and refined in an additional encoding-decoding cycle with the generated inter-block bonds also as inputs. Time values $t$ are randomly sampled at each training step, with the inputting coordinates $\vec{X}^t_i$ interpolated between initial Gaussian-sampled coordinates $\vec{X}^1_i$ and ground-truth coordinates $\vec{X}^0_i$. Detailed training and sampling algorithms are provided in Appendix B. We provide additional discussions and ablation studies on the design choices of the VAE in Appendix C.

### 3.3. Geometric Latent Diffusion Model

The autoencoder simplifies the complicated molecular data, reducing it to compressed latent representations, which are well suited to learn the conditional distribution $p(\mathcal{Z}_x | \mathcal{Z}_y)$ with a diffusion model. Prior to applying the diffusion process, the latent coordinates are normalized by subtracting the center of mass of the binding site latent point cloud $\mathcal{Z}_y$ and divided by an empirical scale factor of 10.0 (Luo et al., 2022). The forward process incrementally adds Gaussian noise to the latent data from $t = 0$ to $t = T$, transitioning the data distribution into the isotropic Gaussian prior $\mathcal{N}(\mathbf{0}, \mathbf{I})$. The reverse process gradually denoises this distribution, reconstructing the original data from $t = T$ to $t = 0$. Denoting the intermediate state of block $i$ at time step $t$ as $\vec{u}^t_i = [z^t_i, \vec{z}^t_i]$, we have the forward process defined as:

$$q(\vec{u}^t_i \mid \vec{u}^{t-1}_i) = \mathcal{N}(\vec{u}^t_i; \sqrt{1 - \beta^t} \cdot \vec{u}^{t-1}_i, \beta^t \mathbf{I}), \quad (11)$$

$$q(\vec{u}^t_i \mid \vec{u}^0_i) = \mathcal{N}(\vec{u}^t_i; \sqrt{\bar{\alpha}^t} \cdot \vec{u}^0_i, (1 - \bar{\alpha}^t)\mathbf{I}), \quad (12)$$

where $\bar{\alpha}^t = \prod_{s=1}^{s=t}(1 - \beta^s)$, and $\beta^t$ is the noise scale conforming to the cosine schedule (Nichol & Dhariwal, 2021). Given the initial state $\vec{u}^0_i$, the state at any time $t$ can be directly sampled as:

$$\vec{u}^t_i = \sqrt{\bar{\alpha}^t} \vec{u}^0_i + \sqrt{1 - \bar{\alpha}^t} \boldsymbol{\epsilon}_i, \quad (13)$$

where $\boldsymbol{\epsilon}_i \sim \mathcal{N}(\mathbf{0}, \mathbf{I})$. The reverse diffusion process reconstructs the data distribution by iteratively removing the noise (Ho et al., 2020), which is modeled as:

$$p_\theta(\vec{u}^{t-1}_i \mid \mathcal{Z}^t_x, \mathcal{Z}_y) = \mathcal{N}(\vec{u}^{t-1}_i; \boldsymbol{\mu}_\theta(\mathcal{Z}^t_x, \mathcal{Z}_y), \beta^t \mathbf{I}), \quad (14)$$

$$\boldsymbol{\mu}_\theta(\mathcal{Z}^t_x, \mathcal{Z}_y) = \frac{(\vec{u}^t_i - \frac{\beta^t}{\sqrt{1-\bar{\alpha}^t}} \boldsymbol{\epsilon}_\theta(\mathcal{Z}^t_x, \mathcal{Z}_y, t)[i])}{\sqrt{\alpha^t}}, \quad (15)$$

where $\alpha^t = 1 - \beta^t$, and $\boldsymbol{\epsilon}_\theta$ is the denoising network also implemented with the equivariant transformer (Jiao et al., 2024), but with only one latent point in each block. The training objective for the diffusion model minimizes the MSE between the predicted noise and the actual noise added during the forward process (Eq. 13), which is given as:

$$\mathcal{L}_{LDM} = \mathbb{E}_{t \sim U(1...T)} \Big[ \frac{\sum_i \|\boldsymbol{\epsilon}_i - \boldsymbol{\epsilon}_\theta(\mathcal{Z}^t_x, \mathcal{Z}_y, t)[i]\|^2}{|\mathcal{Z}^t_x|} \Big], \quad (16)$$

where $|\mathcal{Z}^t_x|$ indicates the number of nodes in the latent point cloud $\mathcal{Z}^t_x$. The overall training and sampling algorithms are included in Appendix D, with discussions on the equivariance in Appendix E. It is worth noting that implementing a diffusion model in the latent space circumvents iterations over full-atom geometries, and thus tremendously reduce the computational cost for training and sampling.

## 4. Experiments

We evaluate our UniMoMo on three categories of molecular binders: peptides, antibodies, and small molecules. For peptides, we use PepBench and ProtFrag datasets (Kong et al., 2024b) with 4,157 protein-peptide complexes and 70,498 synthetic samples for training, and 114 complexes for validation. Testing is conducted on the LNR dataset (Kong et al., 2024b), comprising 93 protein-peptide complexes (Tsaban et al., 2022) with peptide lengths of 4–25 residues. For antibodies, in line with literature (Luo et al., 2022; Kong et al., 2023a), we use SAbDab (Dunbar et al., 2014) entries deposited before September 24th for training and validation, while 60 antigen-antibody complexes from RAbD (Adolf-Bryfogle et al., 2018) are reserved for testing. After filtering for sequence identity above 40% with the test set (Kong et al., 2023a), the training and validation datasets consist of 9,473 and 400 entries, respectively. For small molecules, we adopt Crossdocked2020 and its established splits (Peng et al., 2022), with 99,900 complexes for training and 100 complexes for testing. Additionally, 100 complexes are randomly sampled from the training set for validation.

*Table 1.* Results for *de novo* peptide design.

| Model | Recovery | | | Empirical Energy | | Rationality | | | | Diversity | |
|---|---|---|---|---|---|---|---|---|---|---|---|
| | AAR | C-RMSD | L-RMSD | $\Delta G$ | IMP | Clash$_{in}$ | Clash$_{out}$ | JSD$_{bb}$ | JSD$_{sc}$ | Seq. | Struct. |
| Reference | - | - | - | -37.25 | - | 0.31% | 0.88% | - | - | - | - |
| RFDiffusion | 34.68% | 4.69 | 1.88 | -13.47 | 5.38% | **0.06%** | 13.58% | 0.273 | 0.798 | 0.155 | 0.616 |
| PepFlow | 35.47% | 2.87 | 1.79 | -21.71 | 15.22% | 2.72% | 4.62% | 0.240 | 0.693 | 0.530 | 0.507 |
| PepGLAD | 38.62% | 2.74 | 1.60 | -23.12 | 18.28% | 1.82% | 1.66% | 0.474 | 0.398 | **0.687** | **0.698** |
| UniMoMo (single) | 37.59% | 2.48 | 1.48 | -28.72 | 29.03% | 1.53% | 0.94% | 0.390 | 0.365 | 0.626 | 0.629 |
| UniMoMo (all) | **39.45%** | **2.19** | **1.27** | **-34.35** | **40.86%** | 0.45% | **0.93%** | **0.205** | **0.180** | 0.617 | 0.573 |

For ablation, we include two variants of UniMoMo in the tables throughout this section: one trained exclusively on domain-specific data (**single**) and the other on all datasets (**all**). Note that UniMoMo (**all**) employs the same model weights across benchmarks for different molecular binders, demonstrating its ability to generalize effectively across diverse molecular domains. Other ablations are in Appendix H. In the tables, the best one is marked in bold, with the second best underlined. For small molecules, the third best is also underlined, following the conventions in CBGBench (Lin et al., 2024d).

## 4.1. Peptide

**Setup** We employ the following metrics. **Amino Acid Recovery (AAR)** measures the proportion of generated residues matching the reference peptide. While multiple implementations exist (Luo et al., 2022; Kong et al., 2023a; 2024b; Li et al., 2024b), we adopt the most commonly used one in bioinformatics (Needleman & Wunsch, 1970; Henikoff & Henikoff, 1992), with discussion in Appendix I. As AAR is regarded unreliable (Kong et al., 2024b), we only include it for completeness. **Complex RMSD (C-RMSD)** computes the RMSD of $C_\alpha$ atoms between generated and reference peptides after aligning by the target protein, while **Ligand RMSD (L-RMSD)** is similarly defined with alignment on the peptide itself. $\Delta G$ and **IMP** measure the binding energy calculated by pyRosetta (Alford et al., 2017), and the percentage of target proteins where generated peptides can outperform native binders. **Clash$_{in}$** and **Clash$_{out}$** denote residue-level clashes within the peptide, and between the peptide and the target protein, respectively, defined by $C_\alpha$ atoms below 3.6574Å (Ye et al., 2024). **JSD$_{bb}$** and **JSD$_{sc}$** calculate Jensen-Shannon divergence between dihedral angle distributions of generated peptides and the dataset, for backbone (bb) and sidechain (sc) angles discretized into 10-degree bins (Dunbrack Jr & Cohen, 1997). **Diversity** calculates the ratio of unique clusters to total generations, with sequence identity above 40% and RMSD below 2Å as clustering thresholds. For each target protein in the test set, we generate 100 peptides per model for evaluation.

**Baselines** We adopt the following baselines. **RFDiffusion** (Watson et al., 2023) generates peptide backbones followed by cycles of inverse folding with ProteinMPNN (Dau-

paras et al., 2022) and full-atom relaxation with Rosetta (Alford et al., 2017). For fair comparison, we restrict it to one cycle, as additional cycles optimize Rosetta energy, conferring an unfair advantage over other models in physical metrics. Domain-specific models include **PepFlow** (Li et al., 2024b) and **PepGLAD** (Kong et al., 2024b), which implement multi-modal flow matching and latent diffusion for generating the sequences and the full-atom structures.

**Results** Table 1 demonstrates that our UniMoMo significantly outperforms the baselines in terms of recovery, empirical energy, and rationality, with comparable diversity. Higher recovery metrics (AAR, C-RMSD, and L-RMSD) indicate a greater likelihood of recovering the reference sequences and binding structures. UniMoMo surpasses the baselines by a large margin in pyRosetta binding energy (Alford et al., 2017), reflecting more realistic interaction patterns. Additionally, reduced clash rates and JSD values for dihedral angles demonstrate superior local geometry modeling with higher rationality. Although diversity decreases slightly due to the fidelity-diversity trade-off, it remains comparable to baselines. Notably, the comparison of all-domain UniMoMo with the single-domain UniMoMo reveals clear enhancements across nearly all metrics, particularly binding energy and dihedral angle JSD, highlighting the advantages of leveraging data from multiple domains.

## 4.2. Antibody

**Setup** We define Complementarity-Determining Regions (CDRs) using the Chothia numbering system (Chothia & Lesk, 1987), following Luo et al. (2022). This system, based on structural alignment statistics, offers advantages over the conventional sequence-based IMGT numbering system (Lefranc et al., 2003). Specifically, the IMGT system is vulnerable to simple unigram patterns, which can artificially achieve a high amino acid recovery (AAR) of 39% (Kong et al., 2023b), while the identified pattern only yield 23% AAR under the Chothia system, alleviating this issue. For evaluation, we use conventional metrics (**AAR**, **RMSD**, **IMP**) and rationality metrics for CDRs (**Clash$_{in}$**, **Clash$_{out}$**, **JSD$_{bb}$**, **JSD$_{sc}$**), as previously defined. Moreover, since some baselines adopt predictive implementations (Kong et al., 2023a;b), we also report AAR, RMSD, and IMP across $n$ candidates for each complex, with $n = 1, 10, 100$.

**Baselines** We evaluate our approach against the following baselines. **MEAN**(Kong et al., 2023a) uses a multi-channel equivariant graph neural network with iterative refinement for decoding CDR sequences and structures. **DyMEAN**(Kong et al., 2023b) extends MEAN to full-atom structures and scenarios with unknown antibody framework docking poses. **DiffAb**(Luo et al., 2022) applies diffusion models to amino acid types, $C_\alpha$ coordinates, and residue orientations. **GeoAB**(Lin et al., 2024a) incorporates a heterogeneous residue-level encoder and energy-informed geometric constraints for generating realistic structures.

*Table 2.* Results of recovery for antibody design on CDR-H3.

| Model | #Generation | AAR | RMSD | IMP |
|---|---|---|---|---|
| Predictive | | | | |
| MEAN | 1 | 29.13% | 1.87 | 6.67% |
| dyMEAN | 1 | 31.65% | 8.21 | 11.86% |
| GeoAB-R | 1 | 32.04% | 1.67 | 6.67% |
| Generative | | | | |
| | 1 | 24.60% | 2.77 | 10.34% |
| DiffAb | 10 | 38.42% | 2.08 | 34.48% |
| | 100 | 49.74% | 1.46 | 60.34% |
| | 1 | 29.74% | 1.73 | 6.67% |
| GeoAB-D | 10 | 38.20% | 1.58 | 20.00% |
| | 100 | 45.96% | 1.50 | 40.00% |
| | 1 | 20.44% | 2.71 | 15.00% |
| UniMoMo (single) | 10 | 39.04% | 1.90 | 35.00% |
| | 100 | 48.78% | 1.39 | 63.33% |
| | 1 | 21.44% | 2.52 | 13.33% |
| UniMoMo (all) | 10 | 42.05% | 1.44 | 41.67% |
| | 100 | **52.34%** | **1.04** | **65.00%** |

*Table 3.* Results of rationality for antibody design on CDR-H3.

| Model | Clash$_{in}$ | Clash$_{out}$ | JSD$_{bb}$ | JSD$_{sc}$ |
|---|---|---|---|---|
| Reference | 0.08% | 0.02% | - | - |
| MEAN | 0.96% | 0.16% | 0.529 | - |
| dyMEAN | 1.02% | 2.98% | 0.542 | 0.702 |
| GeoAB-R | 0.59% | 0.11% | 0.529 | - |
| DiffAb | 0.31% | 0.25% | 0.268 | - |
| GeoAB-D | 0.75% | 0.07% | 0.430 | - |
| UniMoMo (single) | 0.25% | 0.06% | 0.278 | 0.284 |
| UniMoMo (all) | **0.18%** | **0.03%** | **0.224** | **0.221** |

**Results** Table 2 demonstrates that our UniMoMo achieves the best performance on generating native-like CDRs. While predictive models show strong single-generation recovery, generative models like UniMoMo demonstrate a clear advantage as the candidate pool increases, achieving higher recovery of the native sequences and structures. With 100 generations, UniMoMo surpasses all baselines on IMP, achieving better binding energy than native CDRs in 65% of complexes. Table 3 further shows that UniMoMo generates realistic geometries with lower clash rates and much more accurate dihedral angle distributions compared to baselines. Notably, compared to the variant of UniMoMo trained on antibody data only, it is evident that UniMoMo can learn transferable knowledge from other domains to improve the quality of the generated interactions and geometries, leading to better recovery and rationality.

## 4.3. Small Molecule

**Setup** We evaluate the results using the CBGBench framework (Lin et al., 2024d), which assesses models across four categories: substructure, chemical property, geometry, and interaction. **Substructure** metrics calculate JSD and MAE between the generated and reference distributions of atoms, rings, and functional groups. **Chemical property** metrics include QED, SA, LogP, and LPSK. **Geometry** metrics focus on JSD for bond lengths and angles, as well as atomic and molecular clash rates. **Interaction** metrics evaluate Vina energy in score, minimization, and dock modes, along with interaction type distributions between molecules and proteins. Models are ranked within each category, with ranks converted to scores using the formula $(N - rank)$, where $N$ is the total number of models. The overall score is a weighted sum of scores across categories, the higher, the better. Detailed descriptions are available in Appendix J.

*Table 4.* Overall comparisons for *de novo* small molecule design.

| Model | substruct. 0.2 | Chem. 0.2 | Interact. 0.4 | Geom. 0.2 | Weighted Score | Rank |
|---|---|---|---|---|---|---|
| LIGAN | 1.13 | 1.40 | **4.27** | 1.25 | 8.05 | 6 |
| 3DSBDD | 1.13 | 1.60 | 2.23 | 0.70 | 5.67 | 9 |
| GraphBP | 0.17 | 1.50 | 0.37 | 0.10 | 2.13 | 14 |
| Pocket2Mol | 0.73 | 1.25 | 2.83 | 0.70 | 5.52 | 10 |
| TargetDiff | 1.77 | 1.50 | 3.50 | 1.70 | 8.47 | 5 |
| DiffSBDD | 0.77 | 1.75 | 1.20 | 0.95 | 4.67 | 12 |
| DiffBP | 0.27 | 1.10 | 2.10 | 1.35 | 4.82 | 11 |
| FLAG | 0.70 | 1.40 | 1.40 | 0.60 | 4.10 | 13 |
| D3FG | 1.47 | **2.25** | 1.80 | 0.70 | 6.22 | 8 |
| DecompDiff | 1.90 | 1.80 | 2.50 | 1.80 | 8.00 | 7 |
| MolCRAFT | 1.93 | 1.55 | 3.93 | **2.20** | 9.62 | 2 |
| VoxBind | 1.53 | 2.00 | 3.83 | 2.00 | 9.37 | 3 |
| UniMoMo (single) | 2.23 | 2.15 | 2.70 | 1.95 | 9.03 | 4 |
| UniMoMo (all) | **2.27** | **2.25** | 3.47 | **2.20** | **10.38** | 1 |

*Table 5.* Substructure analysis for *de novo* small molecule design.

| Model | Atom Type JSD | Atom Type MAE | Ring Type JSD | Ring Type MAE | Functional Group JSD | Functional Group MAE | Rank |
|---|---|---|---|---|---|---|---|
| LIGAN | 0.1167 | 0.8680 | 0.3163 | 0.2701 | 0.2468 | 0.0378 | 8.33 |
| 3DSBDD | 0.0860 | 0.8444 | 0.3188 | 0.2457 | 0.2682 | 0.0494 | 8.33 |
| GraphBP | 0.1642 | 1.2266 | 0.5061 | 0.4382 | 0.6259 | 0.0705 | 13.17 |
| Pocket2Mol | 0.0916 | 1.0497 | 0.3550 | 0.3545 | 0.2961 | 0.0622 | 10.33 |
| TargetDiff | 0.0533 | 0.2399 | 0.2345 | 0.1559 | 0.2876 | 0.0441 | 5.17 |
| DiffSBDD | 0.0529 | 0.6316 | 0.3853 | 0.3437 | 0.5520 | 0.0710 | 10.17 |
| DiffBP | 0.2591 | 1.5491 | 0.4531 | 0.4068 | 0.5346 | 0.0670 | 12.67 |
| FLAG | 0.1032 | 1.7665 | 0.2432 | 0.3370 | 0.3634 | 0.0666 | 10.50 |
| D3FG | 0.0644 | 0.8154 | 0.1869 | 0.2204 | 0.2511 | 0.0516 | 6.67 |
| DecompDiff | 0.0431 | 0.3197 | 0.2431 | 0.2006 | 0.1916 | 0.0318 | 4.50 |
| MolCRAFT | 0.0490 | 0.3208 | 0.2469 | **0.0264** | 0.1196 | 0.0477 | 4.33 |
| VoxBind | 0.0942 | 0.3564 | 0.2401 | 0.0301 | 0.1053 | 0.0761 | 6.33 |
| UiMoMo (single) | 0.0352 | 0.2796 | 0.1252 | 0.0976 | 0.1562 | 0.0173 | 2.83 |
| UniMoMo (all) | **0.0280** | **0.2108** | 0.1073 | 0.0726 | 0.1356 | **0.0161** | **1.67** |

**Baselines** We evaluate our method against a variety of baselines, including autoregressive models such as **Pocket2Mol** (Peng et al., 2022), **GraphBP** (Liu et al., 2022), and **3DSBDD** (Luo et al., 2021). Diffusion-based models, the most prevalent approaches, include **TargetDiff** (Guan et al., 2023a), **DecompDiff** (Guan et al., 2023b), **DiffBP** (Lin et al., 2025), and **DiffSBDD** (Schneuing et al., 2024). Fragment-based methods include **FLAG** (Zhang et al., 2023), which performs one-shot generation, and **D3FG** (Lin et al., 2024a), which auto-regressively generates and connects motifs. Voxel-based methods, which leverage voxel grids and convolutional neural networks, include **LIGAN** (Masuda et al., 2020) and **VoxBind** (Pinheiro

et al., 2024). We further compare against the state-of-the-art model **MolCRAFT** (Qu et al., 2024), which utilizes the Bayesian Flow Network (Graves et al., 2023) for generation.

*Table 6.* Results of chemical properties for *de novo* small molecule design. LogP between -0.4 and 5.6 are all reasonable (Lin et al., 2024d), without preference to higher or lower values.

| Model | QED | LogP | SA | LPSK | Rank |
|---|---|---|---|---|---|
| LIGAN | 0.46 | 0.56 | 0.66 | 4.39 | 7.00 |
| 3DSBDD | 0.48 | 0.47 | 0.63 | 4.72 | 6.00 |
| GraphBP | 0.44 | 3.29 | 0.64 | 4.73 | 6.50 |
| Pocket2Mol | 0.39 | 2.39 | 0.65 | 4.58 | 7.75 |
| TargetDiff | 0.49 | 1.13 | 0.60 | 4.57 | 6.50 |
| DiffSBDD | 0.49 | -0.15 | 0.34 | 4.89 | 5.25 |
| DiffBP | 0.47 | 5.27 | 0.59 | 4.47 | 8.50 |
| FLAG | 0.41 | 0.29 | 0.58 | **4.93** | 7.00 |
| D3FG | 0.49 | 1.56 | 0.66 | 4.84 | **2.75** |
| DecompDiff | 0.49 | 1.22 | 0.66 | 4.40 | 5.00 |
| MolCRAFT | 0.48 | 0.87 | 0.66 | 4.39 | 6.25 |
| VoxBind | 0.54 | 2.22 | 0.65 | 4.70 | 4.00 |
| UniMoMo (single) | 0.53 | 1.36 | 0.68 | 4.69 | 3.25 |
| UniMoMo (all) | **0.55** | 1.55 | **0.70** | 4.68 | **2.75** |

*Table 7.* Geometry analysis for *de novo* small molecule design.

| Model | Static Geometry | | Clash | | Rank |
|---|---|---|---|---|---|
| | $JSD_{BL}$ | $JSD_{BA}$ | $Ratio_{cca}$ | $Ratio_{cm}$ | |
| LIGAN | 0.4645 | 0.5673 | 0.0096 | 0.0718 | 7.75 |
| 3DSBDD | 0.5024 | 0.3904 | 0.2482 | 0.8683 | 10.50 |
| GraphBP | 0.5182 | 0.5645 | 0.8634 | 0.9974 | 13.50 |
| Pocket2Mol | 0.5433 | 0.4922 | 0.0576 | 0.4499 | 10.50 |
| TargetDiff | 0.2659 | 0.3769 | 0.0483 | 0.4920 | 5.50 |
| DiffSBDD | 0.3501 | 0.4588 | 0.1083 | 0.6578 | 9.25 |
| DiffBP | 0.3453 | 0.4621 | 0.0449 | 0.4077 | 7.25 |
| FLAG | 0.4215 | 0.4304 | 0.6777 | 0.9769 | 11.00 |
| D3FG | 0.3727 | 0.4700 | 0.2115 | 0.8571 | 10.50 |
| DecompDiff | 0.2576 | 0.3473 | 0.0462 | 0.5248 | 5.00 |
| MolCRAFT | **0.2250** | **0.2683** | 0.0264 | 0.2691 | **3.00** |
| VoxBind | 0.2701 | 0.3771 | 0.0103 | 0.1890 | 4.00 |
| UniMoMo (single) | 0.3362 | 0.4166 | 0.0042 | 0.0715 | 4.25 |
| UniMoMo (all) | 0.3223 | 0.3848 | **0.0040** | **0.0708** | **3.00** |

**Results** Table 4 presents the overall ranking scores, with detailed evaluations summarized in Tables 5, 6, 7, and 8. UniMoMo achieves the best fidelity to substructure distributions (Table 5), excelling in JSD and MAE for atom types, rings, and functional groups, which is due to the block-level unified representation that effectively captures frequent motifs, such as aromatic rings and functional groups (Kong et al., 2022). The chemical property evaluation in Table 6 reveals the exceptional performance of our UniMoMo in generating drug-like (QED, Bickerton et al. (2012)) molecules with superior synthetic accessibility (SA, Ertl & Schuffenhauer (2009)). The geometry analysis in Table 7 highlights strength of UniMoMo in producing molecular binders with minimal steric clashes and well-distributed bond lengths and angles. In Table 8, UniMoMo exhibits competitive Vina docking scores and interaction profiles closely resembling the reference distribution, suggesting its capability of forming stable and meaningful interactions with target proteins. These results reflect higher structural rationalities, indicating that UniMoMo learns meaningful atom-level geometric constraints with iterative decoding in the full-atom

autoencoder. As shown in Table 4, UniMoMo achieves the highest overall ranking score due to its consistent superiority across the comprehensive evaluation aspects. Importantly, the inclusion of multi-domain data significantly enhances performance in each aspect, indicating that UniMoMo is capable of learning a broader spectrum of property-structure relationships from all types of molecules. This could prevent biases seen in single-domain models, which helps the model generalize better and avoid overfitting to specific chemical spaces, therefore results in a model with better overall performance.

### 4.4. Different Binders for G Protein-Coupled Receptor

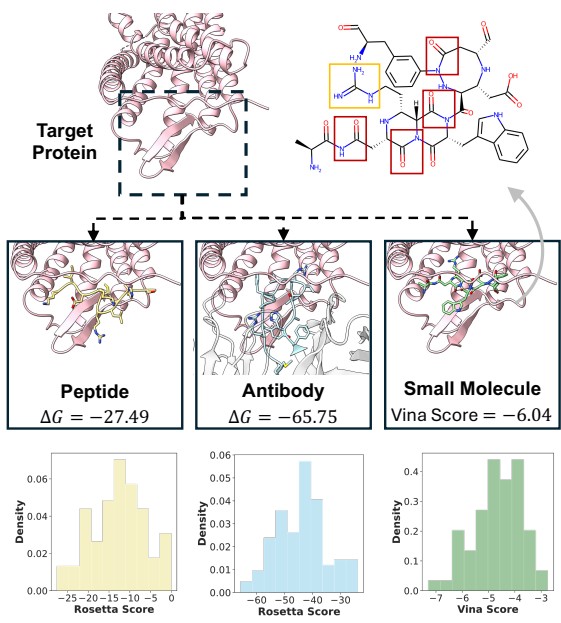

*Figure 3.* Different types of binders generated by UniMoMo on the same binding site on a GPCR (PDB ID: 8U4R). The distribution of Rosetta interface energy is calculated for 100 generated peptides and antibodies. And Vina score is used for evaluation of the 100 generated small molecules. The orange box on the molecular topology graph highlights a structure similar to the side chain of Arginine, while the red boxes denote amide connections.

G protein-coupled receptors (GPCRs), the largest protein family in humans, are among the most prominent drug targets (Yang et al., 2021). We employ UniMoMo to generate peptide, antibody, and small molecule binders targeting the same binding site on a GPCR (PDB ID: 8U4R). Figure 3 shows UniMoMo produces candidates with strong *in silico* binding affinities, assessed by Rosetta (Alford et al., 2017) and Vina (Trott & Olson, 2010), and realistic geometries without requiring relaxation by physical forcefields. The showcased small molecule is particularly interesting, revealing two intriguing phenomena: 1) **Interaction patterns from amino acids**: The small molecule mimics the side chain of Arginine (orange box), one of the standard amino

*Table 8.* Results of interaction analysis for *de novo* small molecule design.

| Model | Vina Score | | Vina Min | | Vina Dock | | | | PLIP Interaction | | | | Rank |
|---|---|---|---|---|---|---|---|---|---|---|---|---|---|
| | E | IMP (%) | E | IMP (%) | E | IMP (%) | MPBG (%) | LBE | $JSD_{OA}$ | $MAE_{OA}$ | $JSD_{PP}$ | $MAE_{PP}$ | |
| LIGAN | **-6.47** | **62.13** | **-7.14** | **70.18** | **-7.70** | **72.71** | 4.22 | 0.3897 | 0.0346 | 0.0905 | 0.1451 | **0.3416** | **3.33** |
| 3DSBDD | - | 3.99 | -3.75 | 17.98 | -6.45 | 31.46 | 9.18 | 0.3839 | 0.0392 | 0.0934 | 0.1733 | 0.4231 | 8.42 |
| GraphBP | - | 0.00 | - | 1.67 | -4.57 | 10.86 | -30.03 | 0.3200 | 0.0462 | 0.1625 | 0.2101 | 0.4835 | 13.08 |
| Pocket2Mol | -5.23 | 31.06 | -6.03 | 38.04 | -7.05 | 48.07 | -0.17 | 0.4115 | 0.0319 | 0.2455 | 0.1535 | 0.4152 | 6.92 |
| TargetDiff | -5.71 | 38.21 | -6.43 | 47.09 | -7.41 | 51.99 | 5.38 | 0.3537 | 0.0198 | 0.0600 | 0.1757 | 0.4687 | 5.25 |
| DiffSBDD | - | 12.67 | -2.15 | 22.24 | -5.53 | 29.76 | -23.51 | 0.2920 | 0.0333 | 0.1461 | 0.1777 | 0.5265 | 11.00 |
| DiffBP | - | 8.60 | - | 19.68 | -7.34 | 49.24 | 6.23 | 0.3481 | 0.0249 | 0.1430 | **0.1256** | 0.5639 | 8.75 |
| FLAG | - | 0.04 | - | 3.44 | -3.65 | 11.78 | -47.64 | 0.3319 | 0.0170 | 0.0277 | 0.2762 | 0.3976 | 10.50 |
| D3FG | - | 3.70 | -2.59 | 11.13 | -6.78 | 28.9 | -8.85 | 0.4009 | 0.0638 | **0.0135** | 0.1850 | 0.4641 | 9.50 |
| DecompDiff | -5.18 | 19.66 | -6.04 | 34.84 | -7.10 | 48.31 | -1.59 | 0.3460 | 0.0215 | 0.0769 | 0.1848 | 0.4369 | 7.75 |
| MolCRAFT | -6.15 | 54.25 | -6.99 | 56.43 | -7.79 | 56.22 | 8.38 | 0.3638 | 0.0214 | 0.0780 | 0.1868 | 0.4574 | 4.17 |
| VoxBind | -6.16 | 41.80 | -6.82 | 50.02 | -7.68 | 52.91 | 9.89 | 0.3588 | 0.0257 | 0.0533 | 0.1850 | 0.4606 | 4.42 |
| UniMoMo (single) | -5.50 | 26.37 | -5.96 | 38.21 | -7.19 | 49.60 | 2.15 | 0.3521 | 0.0175 | 0.0447 | 0.2372 | 0.4706 | 7.25 |
| UniMoMo (all) | -5.72 | 30.40 | -6.08 | 39.23 | -7.25 | 51.59 | 7.50 | 0.3473 | **0.0135** | 0.0173 | 0.2012 | 0.4277 | 5.33 |

acids, to form hydrogen bonds with the target protein. 2) **Local geometries from peptides and antibodies**: To fit in the large pocket originally designed for antibodies, UniMoMo effectively generates wide-spanning scaffolds with amide connections (red boxes) typically seen in peptides and antibodies. The above findings highlight the strong generalization of our UniMoMo across diverse classes of molecular binders, with successful attempts to leverage cross-domain knowledge of molecular interactions and local geometries. Examples on other targets are included in Appendix K. Further analysis reveals an interesting self-adaptive mechanism in block size selection, which depends on the number of blocks designated for generation on the same binding site, with details in Appendix L.

## 5. Conclusion

We introduce UniMoMo, the first unified framework for generative modeling of molecular binders. Molecules are represented as block-based graphs, with each block corresponding to a standard amino acid or a molecular fragment identified via the principal subgraph algorithm. UniMoMo employs a full-atom geometric latent diffusion process, combining an iterative autoencoder to encode blocks as latent points with diffusion modeling in the compressed space. Extensive benchmarks across peptides, antibodies, and small molecules confirm its superiority, illustrating enhanced performance via the integration of data from diverse molecular categories. The application to a GPCR further highlights its ability to transfer interaction patterns and geometries across molecule types. This work is a pioneering step toward unified generative modeling of molecular binders, and may inspire future advancements in this promising direction.

## Code Availability

The codes for data processing, model definition, training, testing, and evaluations are available at https://github.com/kxz18/UniMoMo.

## Acknowlegements

This work is jointly supported by the National Key R&D Program of China (No.2022ZD0160502), the National Natural Science Foundation of China (No. 61925601, No. 62376276, No. 62276152), and Beijing Nova Program (20230484278).

## Impact Statement

Our work presents a novel framework for unified generative modeling of 3D molecules, offering significant value to both academic research and industrial applications. From an academic perspective, we demonstrate that unified modeling across different molecular types is not only feasible but also beneficial, as it enables the model to learn transferable knowledge on molecular interactions and geometries. This proof-of-concept is meaningful for addressing the scientific problem of binder design with computational tools, paving the way for more powerful models that can leverage diverse datasets more effectively. From an industrial standpoint, this work exemplifies the potential to design multiple distinct drugs for the same target protein, addressing varied medical needs by leveraging different molecular formats. In summary, this research has profound implications for accelerating drug discovery and expanding the horizons of molecular binder design. We do not identify any specific societal concerns requiring additional attention at this time.

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

## A. Principal Subgraph Algorithm

Here we borrow descriptions of the principal subgraph algorithm from Kong et al. (2022) to improve the readability of this paper. Algorithm 1 provides the pseudocode for the block-level molecular decomposition with the principal subgraph algorithm. This algorithm takes as input the atom-level molecular graph, a vocabulary of principal subgraphs, and their recorded frequencies of occurrence derived during the principal subgraph extraction process. It iteratively merges two neighboring principal subgraphs whose union has the highest recorded frequency in the vocabulary. This process continues until no unions of two neighboring subgraphs exist in the vocabulary. The function `MergeSubGraph` processes the input molecular graph $\mathcal{G}$ and the selected top-1 fragment $\mathcal{F}$. If $\mathcal{G}$ contains $\mathcal{F}$, the function merges the two adjacent nodes (e.g. $i$ and $j$) in $\mathcal{G}$ that make up $\mathcal{F}$ into a new node representing $\mathcal{F} = \mathcal{F}_i \cup \mathcal{F}_j \cup \mathcal{E}_{ij}$. We further modify the original algorithm to prioritize merging neighboring fragments within the same ring, implemented using the function `RingPrior`. This function identifies all neighboring fragments sharing a ring with the input fragment in the molecule. For any pair of neighboring fragments $i$ and $j$ under consideration for merging, if one fragment has a neighbor in the same ring while the other does not, the pair is discarded to prioritize merging fragments within the same ring. In this paper, we use the vocabulary of 300 principal subgraphs extracted from ChEMBL (Zdrazil et al., 2024) with kekulized representations.

---

**Algorithm 1** Subgraph-Level Decomposition (borrowed and adapted from Kong et al. (2022))

---

1: **Input:** A 2D graph $\mathcal{G}$ decomposed into atoms, the set $\mathbb{V}$ of learned principal subgraphs, and the counter $\mathcal{C}$ of the frequencies for learned principal subgraphs.
2: **Output:** A new representation $\mathcal{G}'$ of $\mathcal{G}$ that consists of principal subgraphs in $\mathbb{V}$.
3: $\mathcal{G}' \leftarrow \mathcal{G}$
4: **while** True **do**
5:     freq $\leftarrow -1$; $\mathcal{F} \leftarrow$ None
6:     **for** $\langle \mathcal{F}_i, \mathcal{F}_j, \mathcal{E}_{ij} \rangle$ **in** $\mathcal{G}'$ **do**
7:         $\mathbb{F}_i, \mathbb{F}_j \leftarrow \mathrm{RingPrior}(\mathcal{F}_i), \mathrm{RingPrior}(\mathcal{F}_j)$   {Get neighboring fragments in the same ring with fragments i and j}
8:         **if** $(\mathcal{F}_j \notin \mathbb{F}_i \text{ and } \mathbb{F}_i \neq \emptyset)$ xor $(\mathcal{F}_i \notin \mathbb{F}_j \text{ and } \mathbb{F}_j \neq \emptyset)$ **then**
9:             continue      {If one of the fragment has a neighbor in a same ring while the other does not, discard this pair}
10:         **end if**
11:         $\mathcal{F}' \leftarrow \mathrm{Merge}(\langle \mathcal{F}_i, \mathcal{F}_j, \mathcal{E}_{ij} \rangle)$                 {Merge neighboring fragments into a new fragment}
12:         $s \leftarrow \mathrm{GraphToSMILES}(\mathcal{F}')$                         {Convert a graph to SMILES representation}
13:         **if** $s \in \mathbb{V}$ and $\mathcal{C}[s] > $ freq **then**
14:             freq $\leftarrow \mathcal{C}[s]$
15:             $\mathcal{F} \leftarrow \mathcal{F}'$
16:         **end if**
17:     **end for**
18:     **if** freq $= -1$ **then**
19:         break                                    {No further merging needed}
20:     **else**
21:         $\mathcal{G}' \leftarrow \mathrm{MergeSubGraph}(\mathcal{G}', \mathcal{F})$                         {Update the graph representation}
22:     **end if**
23: **end while**

---

## B. Algorithms for Training and Sampling with the Iterative Full-Atom Variational Autoencoder

We present the pseudocodes for training and sampling with the iterative full-atom variational autoencoder in Algorithm 2 and 3, respectively. It is worth mentioning that the molecular binder and binding site are encoded independently into latent representations to ensure that the encoded binding site does not include any information on the reference binder. During training, 5% residues of the binding site is sampled for reconstruction, enriching the information encoded into the latent point clouds of the binding site. Ground truth for inter-block bonds is provided with a 50% probability to encourage the model to learn inter-block structural constraints, and facilitate additional refinement cycle with fixed inter-block bonds during inference. The iterative decoder framework allows greater flexibility in controlling molecular geometry. For instance, clash avoidance can be incorporated by introducing a repulsive force on the binding site atoms during each iteration. This framework also offers the potential for fine-grained physical manipulations, such as optimizing directional and angular constraints for hydrogen bonds to achieve more precise interactions, which we leave as a direction for future work.

---

**Algorithm 2** Training Algorithm of the Iterative Full-Atom Autoencoder

---

**input** geometric data of complexes $\mathcal{S}$
**output** encoder $\mathcal{E}_\phi$, decoder $\mathcal{D}_\xi$

1: **function** Encode($\mathcal{E}_\phi, \mathcal{G}$)
2: $\quad \{(\boldsymbol{\mu}_i, \boldsymbol{\sigma}_i, \vec{\boldsymbol{\mu}}_i, \vec{\boldsymbol{\sigma}}_i)\} \leftarrow \mathcal{E}_\phi(\mathcal{G})$ $\hfill$ {Encoding}
3: $\quad \{(\boldsymbol{\epsilon}_i, \vec{\boldsymbol{\epsilon}}_i)\} \sim \mathcal{N}(\boldsymbol{0}, \boldsymbol{I})$ $\hfill$ {Reparameterization}
4: $\quad \mathcal{Z} \leftarrow \{(\boldsymbol{\mu}_i + \boldsymbol{\epsilon}_i \odot \boldsymbol{\sigma}_i, \vec{\boldsymbol{\mu}}_i + \vec{\boldsymbol{\epsilon}}_i \odot \vec{\boldsymbol{\sigma}}_i)\}$
5: $\quad$ **return** $\mathcal{Z}$
6: **end function**
7: Initialize $\mathcal{E}_\phi, \mathcal{D}_\xi$
8: **while** $\phi, \xi$ have not converged **do**
9: $\quad$ Sample $(\mathcal{G}_x, \mathcal{G}_y) \sim \mathcal{S}$
10: $\quad \mathcal{Z}_x, \mathcal{Z}_y \leftarrow \text{Encode}(\mathcal{E}_\phi, \mathcal{G}_x), \text{Encode}(\mathcal{E}_\phi, \mathcal{G}_y)$ $\hfill$ {Encode the molecular binder and the binding site separately}
11: $\quad$ Sample $\tilde{\mathbb{I}}_y \subseteq \mathbb{I}_y$ $\hfill$ {Index of the sampled 5% residues on the binding site for reconstruction}
12: $\quad \mathbb{I} \leftarrow \tilde{\mathbb{I}}_y \cup \mathbb{I}_x$ $\hfill$ {Index of the nodes that need reconstruction}
13: $\quad \{s_i \mid i \in \mathbb{I}\} \leftarrow \mathcal{D}_{\xi_1}(\mathcal{Z}_x, \mathcal{Z}_y)$ $\hfill$ {Block type prediction}
14: $\quad$ Sample $t \sim U(0, 1), \vec{\boldsymbol{X}}_i^1 \sim \mathcal{N}(\vec{\boldsymbol{z}}_i, \boldsymbol{I})$ $\hfill$ {Sample a time value and the initial atomic coordinates at t=1}
15: $\quad \hat{\vec{\boldsymbol{V}}}_i^t \leftarrow \vec{\boldsymbol{X}}_i^0 - \vec{\boldsymbol{X}}_i^1$ $\hfill$ {Ground-truth motion vectors}
16: $\quad \vec{\boldsymbol{X}}_i^t \leftarrow t \cdot \vec{\boldsymbol{X}}_i^1 + (1 - t) \cdot \vec{\boldsymbol{X}}_i^0$ $\hfill$ {Interpolation to get atomic coordinates at t}
17: $\quad \tilde{\boldsymbol{B}}_{ij} \leftarrow \phi$ if $p < 0.5$ else $\hat{\boldsymbol{B}}_{ij}, p \sim U(0, 1)$ $\hfill$ {Provide the ground truth of inter-block bonds with 50% probability}
18: $\quad \{(\boldsymbol{H}_i^t, \vec{\boldsymbol{V}}_i^t) \mid i \in \mathbb{I}\} = \mathcal{D}_{\xi_2}(\{(\hat{\boldsymbol{A}}_i, \hat{\boldsymbol{B}}_i \cup \tilde{\boldsymbol{B}}_{ij}, \vec{\boldsymbol{X}}_i^t)\}, \mathcal{Z}_x, \mathcal{Z}_y, \mathcal{G}_y, t)$ $\hfill$ {Teacher forcing on atoms and bonds}
19: $\quad b_{ij,pq}^t = \text{Softmax}(\text{MLP}(\boldsymbol{h}_{i,p}^t + \boldsymbol{h}_{j,q}^t)), i \neq j$ $\hfill$ {Bond prediction}
20: $\quad \mathcal{L}_{\text{AE}} = \sum_{i \in \mathbb{I}}(\mathcal{L}_{\text{KL}}(i) + \mathcal{L}_{\text{rec}}(i) + \lambda_{\text{dist}}\mathcal{L}_{\text{dist}}(i))/|\mathbb{I}|$ $\hfill$ {Calculate loss}
21: $\quad \phi, \xi \leftarrow \text{optimizer}(\mathcal{L}_{AE}; \phi, \xi)$
22: **end while**
23: **return** $\mathcal{E}_\phi, \mathcal{D}_\xi$

---

**Algorithm 3** Sampling Algorithm of the Iterative Full-Atom Autoencoder

---

**input** encoder $\mathcal{E}_\phi$, decoder $\mathcal{D}_\xi$, binding site $\mathcal{G}_y$, latent point clouds $\mathcal{Z}_x$ and $\mathcal{Z}_y$, number of iterations $N$
**output** molecular binder $\mathcal{G}_x$

1: **function** DecodeStruct($\mathcal{D}_\xi, \{(\boldsymbol{A}_i, \boldsymbol{B}_i, \boldsymbol{B}_{ij})\}, \mathcal{Z}_x, \mathcal{Z}_y, \mathcal{G}_y$)
2: $\quad \Delta t \leftarrow 1.0/N$
3: $\quad \vec{\boldsymbol{X}}_i^0 \sim \mathcal{N}(\vec{\boldsymbol{z}}_i, \boldsymbol{I})$ $\hfill$ {Sample initial coordinates}
4: $\quad$ **for** $n$ in $0, 1, \cdots, N-1$ **do**
5: $\quad\quad t \leftarrow 1.0 - n \cdot \Delta t$
6: $\quad\quad \{(\boldsymbol{H}_i^t, \vec{\boldsymbol{V}}_i^t) \mid i \in \mathbb{I}_x\} \leftarrow \mathcal{D}_{\xi_2}(\{(\boldsymbol{A}_i, \boldsymbol{B}_i \cup \boldsymbol{B}_{ij}, \vec{\boldsymbol{X}}_i^t)\}, \mathcal{Z}_x, \mathcal{Z}_y, \mathcal{G}_y, t)$
7: $\quad\quad \vec{\boldsymbol{X}}_i^{t-\Delta t} \leftarrow \vec{\boldsymbol{X}}_i^t + \Delta t \vec{\boldsymbol{V}}_i^t$ $\hfill$ {Update coordinates}
8: $\quad\quad \boldsymbol{H}_i^{t-\Delta t} \leftarrow \boldsymbol{H}_i^t$
9: $\quad$ **end for**
10: $\quad \boldsymbol{B}_{ij} \leftarrow \{\text{Argmax}(\text{MLP}(\boldsymbol{h}_{i,p}^0 + \boldsymbol{h}_{j,q}^0)) \mid i \neq j, d_{ij,pq}^0 < 3.5\text{Å}\}$ if $\boldsymbol{B}_{ij} = \emptyset$ else $\boldsymbol{B}_{ij}$ $\hfill$ {Inter-block bond prediction}
11: $\quad$ **return** $\{(\vec{\boldsymbol{X}}_i^0, \boldsymbol{B}_{ij}) \mid i \in \mathbb{I}_x\}$
12: **end function**
13: $\{s_i \mid i \in \mathbb{I}_x\} \leftarrow \mathcal{D}_{\xi_1}(\mathcal{Z}_x, \mathcal{Z}_y)$ $\hfill$ {Block type prediction on nodes from the molecular binder}
14: $\boldsymbol{A}_i, \boldsymbol{B}_i \leftarrow \text{Lookup}(s_i, \mathbb{V})$ $\hfill$ {Get atom types and intra-block chemical bonds from the vocabulary}
15: $\{(\vec{\boldsymbol{X}}_i, \boldsymbol{B}_{ij}) \mid i \in \mathbb{I}_x\} \leftarrow \text{DecodeStruct}(\mathcal{D}_\xi, \{(\boldsymbol{A}_i, \boldsymbol{B}_i, \emptyset)\}, \mathcal{Z}_x, \mathcal{Z}_y, \mathcal{G}_y)$ $\hfill$ {Decode coordinates and inter-block bonds}
16: $\mathcal{G}_x \leftarrow (\{(\boldsymbol{A}_i, \vec{\boldsymbol{X}}_i) \mid i \in \mathbb{I}_x\}, \{(\boldsymbol{B}_i, \boldsymbol{B}_{ij}) \mid i \in \mathbb{I}_x, i \neq j\})$
17: $\mathcal{Z}_x \leftarrow \text{Encode}(\mathcal{E}_\phi, \mathcal{G}_x)$ $\hfill$ {Re-encode the generated binder}
18: $\{(\vec{\boldsymbol{X}}_i, \boldsymbol{B}_{ij}) \mid i \in \mathbb{I}_x\} \leftarrow \text{DecodeStruct}(\mathcal{D}_\xi, \{(\boldsymbol{A}_i, \boldsymbol{B}_i, \boldsymbol{B}_{ij})\}, \mathcal{Z}_x, \mathcal{Z}_y, \mathcal{G}_y)$ $\hfill$ {Refinement with given inter-block bonds}
19: $\mathcal{G}_x \leftarrow (\{(\boldsymbol{A}_i, \vec{\boldsymbol{X}}_i) \mid i \in \mathbb{I}_x\}, \{(\boldsymbol{B}_i, \boldsymbol{B}_{ij}) \mid i \in \mathbb{I}_x, i \neq j\})$
20: **return** $\mathcal{G}_x$

## C. Design Choices on the Iterative Full-Atom Variational AutoEncoder

### C.1. Why Do We Need the Compressed Latent Space?

We incorporate a latent space primarily to facilitate the implementation of the diffusion model. In our unified representation, each graph node corresponds to a block, such as a residue or a chemical fragment, whose atom count varies depending on its type. As a result, the block coordinates $\vec{X}_i \in \mathbb{R}^{n_i \times 3}$ and atom types $A_i \in \mathbb{Z}^{n_i}$ are variable in size during denoising, making direct diffusion on these inputs infeasible due to the fixed-dimensional nature of standard diffusion models. To overcome this, we employ an all-atom variational autoencoder (VAE) that encodes each block into fixed-length latent vectors $z_i$ and $\vec{z}_i$, thereby enabling diffusion to operate efficiently and consistently in this fixed-dimensional continuous latent space.

### C.2. Balancing KL Weights on $z_i$ and $\vec{z}_i$

Different KL weights (i.e., $\lambda_1$ and $\lambda_2$ in Eq.2) applied to the invariant ($z_i$) and the equivariant ($\vec{z}_i$) latent variables shape the latent space in distinct ways. To evaluate whether the resulting space is suitable for generative modeling, we use the diffusion loss as a proxy, with a lower loss indicating a better latent space for diffusion. Larger KL weights promote smoother, more continuous latent spaces that benefit diffusion modeling, but excessive regularization may overly compress the representations, diminishing expressivity through information loss. Therefore, these weights must strike a balance to ensure the latent space is both smooth and sufficiently informative. As shown in the validation loss curves in Figure4, we choose the weight combination that yields the lowest validation loss.

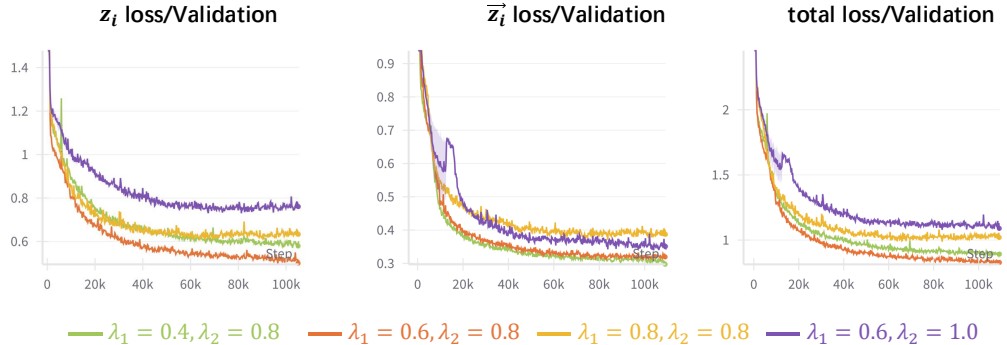

*Figure 4.* Validation loss curves for the latent diffusion model trained on VAE latent spaces with varying KL weights, where $\lambda_1$ and $\lambda_2$ indicate the KL weights on the invariant and the equivariant latent variables, respectively, as in Eq. 2.

### C.3. Ablation Studies on Extra Noises on $\vec{z}_i$

We manually introduce additional noise to the latent coordinates $\vec{z}_i$ before decoding to enhance robustness against errors introduced by the diffusion model. This strategy is particularly beneficial for small molecules, which often exhibit more complex inter-block connectivity than peptides or antibodies, where blocks are typically connected by peptide bonds. These more intricate structures are more sensitive to coordinate perturbations, making robustness especially important. While increasing the KL weight might seem like an alternative way to enforce smoother latent spaces, our current setting of 0.8 for $\vec{z}_i$ already imposes strong regularization. Further increasing it would excessively compress the latent space and degrade the diffusion model's performance, as discussed in Figure 4. As shown in Table 9, adding extra noise significantly improves the tolerance of the autoencoder to coordinate inaccuracies from diffusion, tremendously enhancing stability and affinity of the generated molecules.

*Table 9.* Ablation studies on extra noises and increased KL weights on latent coordinates $\vec{z}_i$.

| model | vina(score only) | vina(minimize) | vina(dock) |
|---|---|---|---|
| w/o extra noise + $\lambda_2 = 0.8$ | -4.62 | -5.58 | -7.24 |
| w/o extra noise + $\lambda_2 = 1.0$ | -4.42 | -5.59 | -7.09 |
| w/ extra noise + $\lambda_2 = 0.8$ | **-5.72** | **-6.08** | **-7.25** |

## D. Algorithms for Training and Sampling with the Latent Diffusion Model

We present the pseudo codes for training and sampling with the latent diffusion model in Algorithm 4 and 5, respectively.

---

**Algorithm 4** Training Algorithm of the Latent Diffusion Model

---

**input** geometric data of complexes $\mathcal{S}$, encoder $\mathcal{E}_\phi$, decoder $\mathcal{D}_\xi$

**output** denoising network $\epsilon_\theta$

 1: Fix parameters $\phi$ and $\xi$
 2: Initialize $\epsilon_\theta$
 3: **while** $\theta$ have not converged **do**
 4:    Sample $(\mathcal{G}_x, \mathcal{G}_y) \sim \mathcal{S}$
 5:    $\mathcal{Z}_x, \mathcal{Z}_y \leftarrow \text{Encode}(\mathcal{E}_\phi, \mathcal{G}_x), \text{Encode}(\mathcal{E}_\phi, \mathcal{G}_y)$           {Encoding}
 6:    $t \sim \mathbf{U}(1, T)$, $\{(\epsilon_i, \vec{\epsilon}_i)\} \sim \mathcal{N}(\mathbf{0}, \boldsymbol{I})$           {Sample noises}
 7:    $\mathcal{Z}_x^t \leftarrow \{(z_i^t, \vec{z}_i^t) \mid i \in \mathbb{I}_x, [z_i^t, \vec{z}_i^t] = \sqrt{\bar{\alpha}^t}[z_i^0, \vec{z}_i^0] + (1 - \bar{\alpha}^t)[\epsilon_i, \vec{\epsilon}_i]\}$   {Sample intermediate states at time t}
 8:    $\mathcal{L}_{LDM}^t = \sum_i \|[\epsilon_i, \vec{\epsilon}_i] - \epsilon_\theta(\mathcal{Z}_x^t, \mathcal{Z}_y, t)[i]\|^2 / |\mathcal{Z}_x^t|$
 9:    $\theta \leftarrow \text{optimizer}(\mathcal{L}_{LDM}^t; \theta)$
10: **end while**
11: **return** $\epsilon_\theta$

---

**Algorithm 5** Sampling Algorithm of the Latent Diffusion Model

---

**input** encoder $\mathcal{E}_\phi$, decoder $\mathcal{D}_\xi$, denoising network $\epsilon_\theta$, binding site $\mathcal{G}_y$, diffusion steps $T$, decoding iterations $N$

**output** molecular binder $\mathcal{G}_x$

 1: $\mathcal{Z}_y \leftarrow \text{Encode}(\mathcal{E}_\phi, \mathcal{G}_y)$           {Encode the binding site into the latent space}
 2: $\{(z_i^T, \vec{z}_i^T)\} \sim \mathcal{N}(\mathbf{0}, \boldsymbol{I})$           {Sample initial states for diffusion}
 3: **for** $t$ in $T, T-1, \cdots, 1$ **do**
 4:    $\mathcal{Z}_x^t \leftarrow \{(z_i^t, \vec{z}_i^t) \mid i \in \mathbb{I}_x\}$           {Latent Denoising Loop for nodes in the molecular binder}
 5:    $\vec{u}_i^t \leftarrow [z_i^t, \vec{z}_i^t]$
 6:    $\varepsilon_i = [\epsilon_i, \vec{\epsilon}_i] \sim \mathcal{N}(\mathbf{0}, \boldsymbol{I})$
 7:    $\vec{u}_i^{t-1} \leftarrow \frac{1}{\sqrt{\alpha^t}}(\vec{u}_i^t - \frac{\beta^t}{\sqrt{1-\bar{\alpha}^t}}\epsilon_\theta(\mathcal{Z}_x^t, \mathcal{Z}_y, t)[i]) + \beta_t \varepsilon_i$    {Update the latent point cloud of the molecular binder}
 8:    $[z_i^{t-1}, \vec{z}_i^{t-1}] \leftarrow \vec{u}_i^{t-1}$
 9: **end for**
10: $\mathcal{Z}_x \leftarrow \{(z_i^0, \vec{z}_i^0) \mid i \in \mathbb{I}_x\}$
11: $\mathcal{G}_x \leftarrow \text{Decode}(\mathcal{E}_\phi, \mathcal{D}_\xi, \mathcal{G}_y, \mathcal{Z}_x, \mathcal{Z}_y, N)$           {Iterative decoding in Algorithm 3}
12: **return** $\mathcal{G}_x$

---

## E. Discussion on E(3)-Equivariance

We briefly outline how E(3)-equivariance is preserved throughout our framework, building upon established theoretical foundations.

For the diffusion component, GeoDiff (Xu et al., 2022) demonstrates that E(3)-equivariance is maintained via an equivariant denoising kernel (Proposition 1), which in our paper is instantiated using an equivariant transformer (Jiao et al., 2024). This transformer operates on scalar features, preserving equivariance through inner products and unbiased linear transformations applied to velocity vectors, in accordance with the scalar-based design principles Villar et al. (2021).

For the variational autoencoder, the encoder outputs latent variables using the same equivariant transformer, ensuring E(3)-equivariance by design as the latents are the direct output of the transformer. The decoder adopts a short flow matching approach, with a similar theoretical foundation as GeoDiff on equivariant flow matching (Song et al., 2023) (Theorem 4.1). Since the vector field is predicted by an E(3)-equivariant network, the decoding process consistently maintains E(3)-equivariance across all stages.

# F. Implementation Details

## F.1. Baselines

**Peptide**    For RFDiffusion, as the training codes for custom datasets are not open-source, we directly utilize the official pretrained weights for binder design and the provided inference framework. The number of cycles for inverse folding and Rosetta relaxation is set to one. For PepGLAD and PepFlow, we use the official implementations and retrain the models on the same datasets as our single-domain model, with the default hyperparameters specified in their repositories.

**Antibody**    For all baseline methods, including MEAN, dyMEAN, DiffAb, GeoAB-R, and GeoAB-D, we employ their official implementations to retrain the models on the same datasets as our single-domain model, using the default hyperparameters provided in their repositories.

**Small Molecule**    The baseline results are borrowed from CBGBench (Lin et al., 2024d), where the models are retrained on the same datasets and splits as our single-domain model.

## F.2. UniMoMo

We provide the hyperparameters for training our single-domain and multi-domain models in Table 10.

*Table 10.* Hyperparameters of UniMoMo with single-domain and multi-domain data.

| Name | Value | | Description |
| --- | --- | --- | --- |
| | UniMoMo (single) | UniMoMo (all) | |
| Variational AutoEncoder | | | |
| epoch | 250 | 250 | Number of epochs to train. |
| warmup | 2000 | 2000 | Number of warmup steps for the $\mathcal{L}_{KL}$, with linear schedule. |
| lr | 1e-4 | 1e-4 | Learning rate. |
| embed_size | 512 | 512 | Dimension of the atom and block type embeddings. |
| hidden_size | 512 | 512 | Dimension of hidden states. |
| latent_size | 8 | 8 | Dimension of latent states. |
| edge_size | 64 | 64 | Dimension of edge type embeddings. |
| n_rbf | 64 | 64 | Number of RBF kernels for embedding the spatial distances. |
| cutoff | 10.0 | 10.0 | Cutoff distance for RBF kernels. |
| n_layers | 6 | 6 | Number of layers in the encoder and the decoder. |
| n_head | 8 | 8 | Number of heads for multi-head attention. |
| $\lambda_1$ | 0.4 (small molecule) 0.6 (others) | 0.6 | The weight of KL divergence on the sequence. |
| $\lambda_2$ | 0.8 | 0.8 | The weight of KL divergence on the structure. |
| $\lambda_{dist}$ | 0.5 | 0.5 | The weight of local distance loss. |
| mask_ratio | 0.05 | 0.05 | The ratio of residues on the binding site for reconstruction. |
| N | 10 | 10 | Number of iterations for reconstruction. |
| Latent Diffusion Model | | | |
| hidden_size | 512 | 512 | Dimension of hidden states. |
| T | 100 | 100 | Number of total steps for diffusion. |
| n_rbf | 64 | 64 | Number of RBF kernels for embedding the spatial distances. |
| cutoff | 3.0 | 3.0 | Cutoff distance for RBF kernels in the normalized latent space. |
| n_layers | 6 | 6 | Number of layers in the denoising network. |
| n_head | 8 | 8 | Number of heads for multi-head attention. |

# G. Physical Corrections on Generation

To enhance the physical plausibility of generated structures, different constraints could be incorporated into the proposed framework. Here we illustrate two common strategies to correct the clashes and the local geometries.

### G.1. Clashes Avoidance

Clashes frequently occur in structures generated by deep learning models. In UniMoMo, we decouple coarse global arrangement and fine-grained local geometry using latent diffusion and a full-atom variational autoencoder, which allows us to explicitly handle steric clashes during decoding by adding repulsive forces to the predicted vector field (Eq.5) during iterative reconstruction. At each step, we compute pairwise distances between generated atoms and those in the contexts (i.e. the binding site), and apply a force when their van der Waals radii overlap beyond a threshold $\delta$ (Ramachandran et al., 2011), which we use 0.3Å in this paper. Specifically, if $r_i$ and $r_j$ are the radii of atoms $i$ and $j$, and $\vec{x}_i^t$ and $\vec{x}_j^t$ are their coordinates at time step $t$, the magnitude of the repulsive force is defined as:

$$f_{ij} = \begin{cases} r_i + r_j - \delta - \|\vec{x}_i^t - \vec{x}_j^t\|, & \|\vec{x}_i^t - \vec{x}_j^t\| < r_i + r_j - \delta, \\ 0, & else, \end{cases} \tag{17}$$

Then for each iteration during decoding, the clashes can be repaired by further updating the atomic coordinates as $\vec{x}_i^t = \vec{x}_i^t + \sum_j \frac{\vec{x}_i^t - \vec{x}_j^t}{\|\vec{x}_i^t - \vec{x}_j^t\|} \cdot f_{ij}$, where $j$ iterates over all atoms in the fixed context.

### G.2. Bond Validity

Existing literature often relies on OpenBabel (O'Boyle et al., 2011) to assign chemical bonds based on pairwise atomic distances. In contrast, our unified framework determines bonding through block-defined intra-block bonds and model-predicted inter-block bonds. While this model-driven approach allows greater flexibility, such as accommodating non-standard amino acids for future extension, it may introduce issues like invalid valencies and inconsistencies between predicted 2D topology and 3D geometry (e.g., excessively long bond lengths or distorted bond angles). To address valency violations, we rank predicted bonds by their output probability and add them sequentially. Any bond that exceeds valency constraints is discarded. For 2D–3D consistency, we compute empirical distributions of bond lengths and angles, with bond lengths discretized from 1.1Å to 1.7Å (in 0.005Å bins), and angles from 0° to 180° (in 2° bins). If over 10% of the bonds in a generated molecule fall into zero-probability regions of these distributions, the molecule is considered 2D-3D inconsistent and discarded.

## H. Ablations on the Number of Iterations

We conduct an ablation study on the iterative decoding module, which primarily impacts the physical validity of local geometries, including bond lengths, bond angles, and dihedral angles. As shown in Table 11, reducing the number of iterations significantly degrades the precision of atom-level geometries, resulting in substantial divergence from natural distributions. Notably, when $N = 1$, the decoding process becomes non-iterative, leading to a tremendous drop in performance compared to its iterative counterparts.

*Table 11.* Ablations on the effect of the number of iterations in the decoder.

| $N$ | Peptide | | Antibody | | Small Molecule | |
|---|---|---|---|---|---|---|
| | $JSD_{bb}$ | $JSD_{sc}$ | $JSD_{bb}$ | $JSD_{sc}$ | $JSD_{BL}$ | $JSD_{BA}$ |
| 1 | 0.393 | 0.358 | 0.376 | 0.374 | 0.4216 | 0.5328 |
| 3 | 0.281 | 0.232 | 0.277 | 0.260 | 0.3625 | 0.4976 |
| 5 | 0.273 | 0.203 | 0.270 | 0.242 | 0.3470 | 0.4724 |
| 10 | **0.205** | **0.180** | **0.224** | **0.221** | **0.3223** | **0.3848** |

## I. Implementation of Amino Acid Recovery (AAR)

We found that various implementations of AAR exist in the current literature, which we illustrate as follows. DiffAb (Luo et al., 2022) calculates AAR using a global pairwise alignment of two sequences with the BLOSUM62 matrix (Henikoff & Henikoff, 1992) and the Needleman-Wunsch algorithm (Needleman & Wunsch, 1970), as implemented in Biopython (Cock et al., 2009). It then computes the ratio of matched amino acids between the generated and reference sequences.

MEAN (Kong et al., 2023a) uses a simpler approach, directly matching amino acids from the start of both sequences without alignment. PepGLAD (Kong et al., 2024b) assumes that sequence alignment for peptides should not necessarily start at the beginning; for instance, the AAR of `ARGFE` to `RGFED` would be 80% rather than 0%. Therefore, it implements a sliding window approach, taking the maximum match ratio as the result. PepFlow (Li et al., 2024b) utilizes `SequenceMatcher` from the `difflib` package in python, which employs the Ratcliff/Obershelp algorithm. This method recursively identifies the longest common substring (LCS) on either side of the previous LCS. For example, comparing `DIET` and `TIDE` results in an AAR of 50% by first identifying `I` as the initial LCS, followed by `E` as a secondary LCS on the right side of the previous LCS. This often leads to much higher AAR values than other methods.

Given the biological relevance of the BLOSUM62 matrix (Henikoff & Henikoff, 1992) and the Needleman-Wunsch algorithm (Needleman & Wunsch, 1970), we adopt the implementation of DiffAb to calculate AAR in our study.

## J. Descriptions of Metrics in CBGBench

The evaluation of CBGBench (Lin et al., 2024d) involves the following four aspects: substructure, chemical property, geometry, and interaction.

**Substructure** The comparison focuses on **atom types**, **ring types**, and **functional groups**, assessing how closely the generated distributions align with the reference. Two types of metrics are involved: Jensen-Shannon divergence (JSD), and mean absolute error (MAE). **JSD**($\downarrow$) calculates the divergence between the overall probablistic distributions on the specific substructures of the generated and the reference molecules. **MAE**($\downarrow$) calculates the difference between the molecule-level occurring frequencies of substructures (e.g. carbon atoms occur 16 times in one molecule on average) of the generated and reference molecules. Atom types considered include `C`, `N`, `O`, `F`, `P`, `S`, `Cl`, while ring types range from 3 to 8 atoms. Functional groups are evaluated based on 25 categories identified using the `EFG` algorithm (Salmina et al., 2015).

**Chemical Property** The evaluation of chemical properties builds on prior work, incorporating several key metrics. These include **QED**($\uparrow$), which provides a quantitative estimation of drug-likeness (Bickerton et al., 2012); **SA**($\uparrow$), the synthetic accessibility score (Ertl & Schuffenhauer, 2009) normalized to $[0, 1]$, with higher scores indicating better synthesizability; and **LogP**(-), the octanol-water partition coefficient, with optimal values typically ranging between -0.4 and 5.6 for drug candidates (Ghose et al., 1999). Additionally, the **LPSK**($\uparrow$) metric measures the proportion of generated drug molecules that satisfy Lipinski's rule of five (Lipinski, 2004), a guideline for assessing drug-like properties.

**Geometry** This panel assesses the authenticity of generated molecules by analyzing local geometries, including bond lengths, bond angles, and atomic clashes. **JSD$_{BL}$**($\downarrow$) quantifies the Jensen-Shannon divergence between the bond length distributions of generated and reference molecules. Similarly, **JSD$_{BA}$**($\downarrow$) evaluates the divergence for bond angles, which are discretized every two degrees between $0°$ to $180°$. Clashes between generated molecules and target proteins are identified based on van der Waals radius overlaps of $\geq 0.4$Å (Ramachandran et al., 2011). **Ratio$_{cca}$**($\downarrow$) measures the average proportion of clashing atoms, while **Ratio$_{cm}$**($\downarrow$) reflects the average proportion of molecules with at least one clashing atom.

**Interaction** The evaluation of interactions encompasses two primary aspects: Vina scores and interaction distributions. For the Vina score analysis, three modes of `AutoDock Vina` (Trott & Olson, 2010) are utilized: **Score**, which directly calculates the energy for the generated binding conformation; **Min**, which optimizes the docking pose of the generated molecule while keeping its internal geometry fixed; and **Dock**, which simultaneously optimizes both the docking pose and internal geometry. For each mode, **E**($\downarrow$) represents the average Vina energy, while **IMP(%)**($\uparrow$) calculates the percentage of generated molecules with better Vina energy than the reference molecules. Two metrics are further introduced to reduce the bias of Vina energy on the size of molecules: mean percent binding gap (MPBG) and ligand binding efficiency (LBE). **MPBG**($\uparrow$) measures relative improvement of generated molecules over reference molecules in terms of Vina energy for each binding pocket, defined as $\text{MPBG} = \sum_i(\sum_j \frac{E_{ij,\text{gen}} - E_{i,\text{ref}}}{N \cdot E_{i,\text{ref}}})$. **LBE**($\uparrow$) evaluates the average energy contribution per atom, calculated as $\text{LBE} = -E/N_{atom}$. Interaction distribution metrics assess the similarity between generated and reference molecules across seven interaction types identified by `PLIP` (Salentin et al., 2015). **JSD$_{OA}$**($\downarrow$) and **MAE$_{OA}$**($\downarrow$) evaluate overall distributions of interaction types, while **JSD$_{PP}$**($\downarrow$) and **MAE$_{PP}$**($\downarrow$) focus on per-pocket distributions.

## K. Case Study

In Figure 5 and 6, we provide additional designs of peptides, antibodies, and small molecules on four targets: SARS receptor binding domain, cluster of differentiation 38, Influenza H5 HA head domain, and HIV envelope glycoprotein gp160.

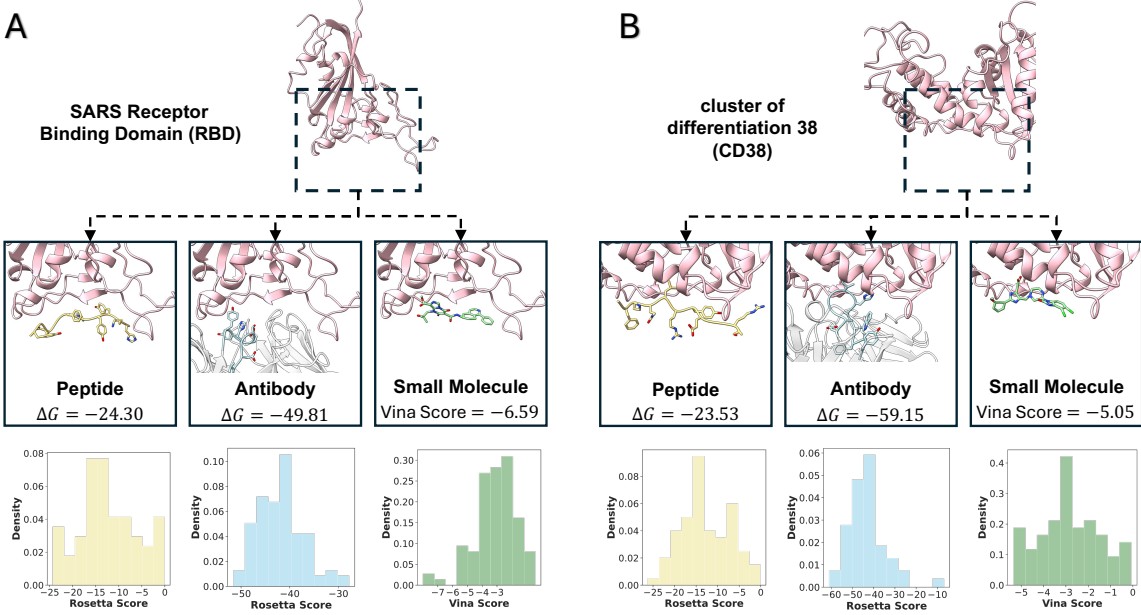

*Figure 5.* Designed peptides, antibodies, and small molecules, as well as their *in silico* binding affinity distributions, for the same binding site on **(A)** SARS receptor binding domain (PDB ID: 2GHW) and **(B)** cluster of differentiation 38 (PDB ID: 4CMH).

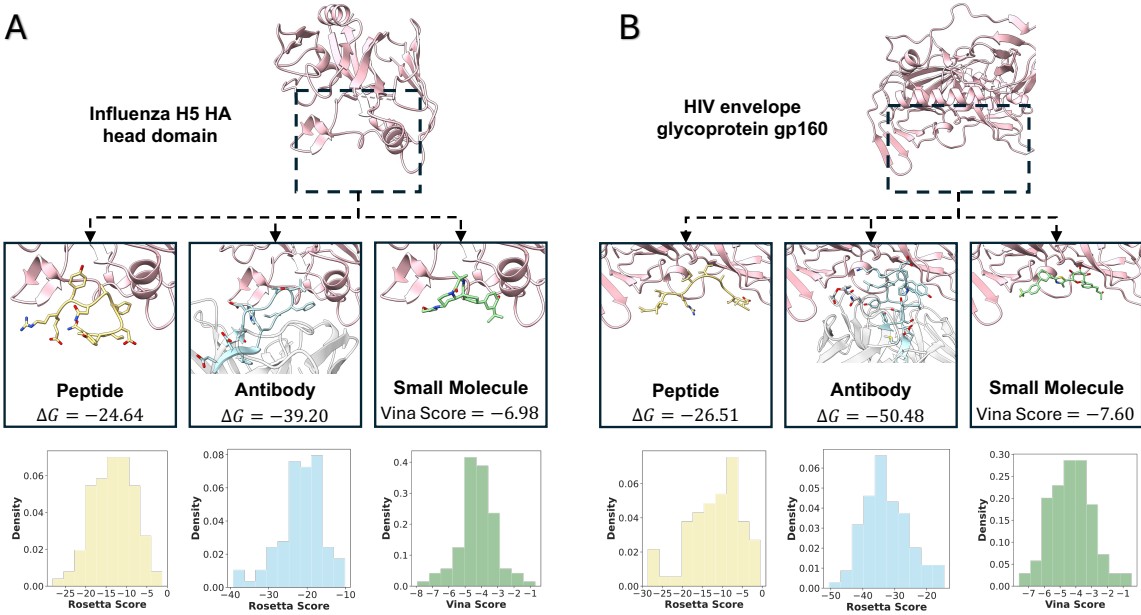

*Figure 6.* Designed peptides, antibodies, and small molecules, as well as their *in silico* binding affinity distributions, for the same binding site on **(A)** Influenza H5 HA head domain (PDB ID: 4XNQ) and **(B)** HIV envelope glycoprotein gp160 (PDB ID: 4YDK).

## L. Self-Adaptive Molecular Sizes

We investigate how the choices of number of blocks influence generations on the same binding site. Using the GPCR case from § 4.4, we evaluate different block configurations averaged over 100 generations. For peptides, we vary the number of

blocks into three categories: small (4–10 blocks), medium (11–17 blocks), and large (18–24 blocks). For small molecules, the original algorithm follows the standard protocol of sampling the number of blocks based on the spatial size of the binding site (Guan et al., 2023a). We modify this by shifting the distribution $\pm 5$ blocks to assign less or more number of blocks. As shown in Table 12, peptide results align with expectations, where larger binders generally yield lower binding energies by forming more interactions. Remarkably, for small molecules, the model exhibits a self-adaptive behavior. When fewer blocks are used, it generates larger fragments with more atoms to fill the pocket; conversely, with more blocks, it opts for smaller fragments to avoid steric clashes, indicating context-aware flexibility in generation.

*Table 12.* Comparisons of energies and number of atoms in each block for peptides and small molecules under different settings of designated number of blocks.

| Peptide | | Small Molecule | | |
|---|---|---|---|---|
| #Blocks | Rosetta dG | #Blocks | Vina Score (dock) | Avg. #Atoms per Block |
| small (4-10) | -8.09 | small (n-5) | -6.72 | 5.10 |
| medium (11-17) | -16.93 | medium (n) | -7.42 | 4.06 |
| big (18-24) | -22.39 | big (n+5) | -7.44 | 3.35 |

