# OpenReview forum: "UniMoMo: Unified Generative Modeling of 3D Molecules for De Novo Binder Design"
_ICML.cc/2025/Conference — ICML 2025 poster_

### Official Review · Reviewer_gQC8 · 2025-03-12

**Overall Recommendation:** 3

**Summary:**

This paper introduces a unified framework, called UniMoMo, for general target-specific binder generation. The target is a protein, and the binders could be peptides, antibodies, or small molecules. UniMoMo aims to train a single generative model to tackle general binder generation problem, while being able to leverage datasets across different domains. The performance on various binder generation benchmarks has been demonstrated.

**Claims And Evidence:**

The main claim is that by using a unified generative model, we can tackle different binder design task at once. Also, the dataset from different binder domains can help each other. This has been verified by the great experimental performance across several widely used benchmarks.

**Essential References Not Discussed:**

None

**Experimental Designs Or Analyses:**

All the experimental designs are closely following the community standard, and they are solid to me.

**Methods And Evaluation Criteria:**

The main contributions of this unified framework are the unified representation of graph of blocks and the geometric latent diffusion model. The overall UniMoMo framework is sound. In terms of evaluation, the authors conducted experiments on widely used benchmarks for target-conditioned peptide generation, antibody generation, and small molecule generation. The performance is strong across benchmarks compared to existing methods that are specifically designed for each domain. Also, it shows that leveraging datasets across domain with the UniMoMo framework is helpful for boosting performance.

**Other Comments Or Suggestions:**

None

**Other Strengths And Weaknesses:**

None

**Questions For Authors:**

How did you balance data from different all three domains during training?

**Relation To Broader Scientific Literature:**

The proposed UniMoMo is unique as it unifies binder design for different binder types. Moreover, it shows training on the combined datasets from all domains can help improve performance for each domain, which is exciting.

**Theoretical Claims:**

N/A. No major theoretical claims.

---

> ### Author Rebuttal · Authors · 2025-03-31
>
> Thanks for your appreciation and the positive comments!
>
> > Q1: How did you balance data from different all three domains during training?
>
> Thanks for the question! In the current implementation, we have not extensively explored domain-specific data balancing strategies. For our joint training approach, we utilized representative datasets from each domain: CrossDocked (\~100k samples) for small molecules, PepBench (\~80k samples) for peptides, and SAbDab (\~10k samples) for antibodies. As an initial effort toward unified molecular modeling, we employed simple random sampling across these datasets during training, which has demonstrated promising results. Moving forward, particularly as we scale the framework to incorporate larger and more diverse datasets, we recognize the importance of investigating more sophisticated data balancing approaches to further enhance model performance.

---

### Official Review · Reviewer_gwNY · 2025-03-14

**Overall Recommendation:** 4

**Summary:**

In this paper, the authors propose a new generative model for 3D molecule design conditioned on a protein target.
The proposed model, UniMoMo, unifies generation of different ligand modalities (small molecules, peptides and parts of antibodies) into a single model. This is done by considering each molecule, independent of their modalities, as a graph of blocks (amino acids for peptides/antibodies or molecular substructures for molecules).
The proposed approach is a latent generative model with three parts: (i) a autoencdoer that encodes blocks into a latent space then decode the back to block types, (ii) a latent diffusion model operating on the learned latent space, and (iii) an iterative generation approach to recode full-atom geometries from the blocks.
The authors show good results on three benchmarks (for three different molecular modalities). More interestingly, they show that training a model on all modalities is usually better than training each modality independently.


## Update after rebuttal
I thank the authors for their rebuttal. I will keep my score with the requirement that the authors update the manuscript to make the points fellow reviewers and I pointed out, specially when it comes to better details and explanations.
Congratulations!

**Claims And Evidence:**

Yes

**Essential References Not Discussed:**

Not that I am aware of.

**Experimental Designs Or Analyses:**

- The paper only evaluates on in-silico metrics, which are known to be far from perfect. The proposed method do achieve good results on these metrics (compared to baselines benchmarked).
- I think a lot of experimental details are missing, specially when it comes to the antibody and peptides experiments. For example, which parts of the CDR is modelled? The loops? The frames? Everything? How many AAs are considered on this setting?
- During sampling time, how many "block" nodes are chosen before starting the (reverse) diffusion process? How does this choice affects experimental results (on all modalities)?
- During sampling time, given a tartget pocket, how does the model decide when it generates blocks that belond to small molecules, or peptides or antibodies?

**Methods And Evaluation Criteria:**

Yes. The authors propose a model that is modality-agnostic and show results on three different molecular modalities.

**Other Comments Or Suggestions:**

More details on pepdtides and antibodies experimental section is neeeded.

**Other Strengths And Weaknesses:**

Strengths:
- This is one of the first approaches that proposes a unified model for structure-based molecule design that can be applied to different molecular modalities. Moreover, the authors show that training on all modalities is better than training on a single modality.

Weakness:
- The proposed approach contains many different sub-components, making it hard to reproduce/build upon, and probably reducing its effectiveness and expressivity.

**Questions For Authors:**

- See above for more questions.
- During sampling time, how many "blocks" nodes are chosen before starting the (reverse) diffusion process? How does this choice affects experimental results (on all modalities)?
- I feel some information is missing on how the bonds are computed. Could the authors elaborate more on how the bonds between bonds and between blocks are computed? It seems that the latter depends on a NN prediction. How accurate is it?
- How are the sequences of AAs (on both peptides and antibodies) extracted from the full atom point cloud?
- What parts of the CDR are modelled on the Antibody experiments? More experimental details on this section is needed.
- During sampling time, given a target pocket, how does the model decide when it generates blocks that belond to small molecules, or peptides or antibodies?

**Relation To Broader Scientific Literature:**

The paper is well-placed in the context of structure-conditioned 3D molecule generation. They provide a method that model all-atoms and can be applied on different data molecule modalities.

**Theoretical Claims:**

N/A

---

> ### Author Rebuttal · Authors · 2025-03-31
>
> Thanks for your appreciation and the constructive comments!
>
> > Q1: The paper only evaluates on in-silico metrics, which are known to be far from perfect. The proposed method do achieve good results on these metrics (compared to baselines benchmarked).
>
> Thanks for the comments! While we are currently conducting wet-lab validations, the experimental timeline is extensive. Therefore, we plan to include these results in a future extension of this work.
>
> > Q2: During sampling time, how many "blocks" nodes are chosen before starting the (reverse) diffusion process? How does this choice affects experimental results (on all modalities)?
>
> For peptides and antibodies, we follow the literature [a,b] and set the number of blocks equal to that of the reference sequence, as many metrics, including AAR and RMSD, assume generated sequences having the same lengths as the native binders.
>
> For small molecules, the number of blocks is sampled from the statistical distribution based on the spatial size of the pocket [c]. We evaluated different settings on the GPCR case in Section 4.4, averaging results over 100 designs, where $n$ denotes the originally sampled number of blocks for small molecules:
>
> | peptide lengths | Rosetta dG |
> |-|-|
> | small(4-10)     | -8.09   |
> | medium(11-17)  | -16.93  |
> | big(18-24)    | -22.39  |
>
> | molecule fragments | Vina score (dock) | Avg.atom per block |
> | -------- | --------| -- |
> | small (n-5)  | -6.72   | 5.10 |
> | medium (n)   | -7.42  | 4.06 |
> | big (n+5)   | -7.44  | 3.35 |
>
> Commonly, larger binders lead to lower energies, since they can form more interactions. Interestingly, for small molecules, the model adapts to the number of blocks. With fewer blocks, it generates larger fragments with more atoms to fill the pocket; otherwise, with more blocks, it generates smaller fragments, avoiding overcrowding the pocket.
>
> Thanks for the insightful question again! We will include the discussion in the revision!
>
> [a] Full-Atom Peptide Design based on Multi-modal Flow Matching. ICML 2024.
>
> [b] Conditional Antibody Design as 3D Equivariant Graph Translation. ICLR 2023.
>
> [c] CBGBench: Fill in the Blank of Protein-Molecule Complex Binding Graph. ICLR 2025.
>
> > Q3: I feel some information is missing on how the bonds are computed. Could the authors elaborate more on how the bonds between bonds and between blocks are computed? It seems that the latter depends on a NN prediction. How accurate is it?
>
> Sorry for the confusion. For intra-block bonds, they are predetermined once the block type is assigned. For inter-block chemical bonds, we employ an MLP for prediction, using the hidden states of atom pairs as input (Eq. 6). Bond prediction is restricted to spatially neighboring atoms, excluding those that are too far apart. The predictions are dynamically changing during the flow matching process in the decoder (Figure 1B, bottom right). Ultimately, the reconstruction accuracy of the chemical bonds are around 97%, which we think is accurate enough.
>
> > Q4: How are the sequences of AAs (on both peptides and antibodies) extracted from the full atom point cloud?
>
> Sorry for the confusion. In our block-level decomposition, each natural amino acid forms a block. Therefore, we can directly get the amino acid type of each block. We think this is also one merit of our block-based unified representation, without requirements to further derive an algorithm to extract the AAs.
>
> > Q5: What parts of the CDR are modelled on the Antibody experiments? More experimental details on this section is needed.
>
> Sorry for the confusion. We follow the convention [d, e] and evaluate on CDR-H3, as it exhibits much higher irregularities than other CDRs and plays a crucial role in binding and interactions [f]. We appreciate the suggestion and will clarify this in the revision.
>
> [d] End-to-End Full-Atom Antibody Design. ICML 2023.
>
> [e] GeoAB: Towards Realistic Antibody Design and Reliable Affinity Maturation. ICML 2024.
>
> [f] Antigen-Specific Antibody Design and Optimization with Diffusion-Based Generative Models for Protein Structures. NeurIPS 2022.
>
> > Q6: During sampling time, given a target pocket, how does the model decide when it generates blocks that belond to small molecules, or peptides or antibodies?
>
> Thanks for the insightful question! We apologize for not making this clear in the paper. We use a binary prompt for each block, with 1 indicating the generation of amino acid (AA), and 0 indicating either AAs or molecular fragments. During benchmarking for peptides and antibodies, all blocks are assigned a prompt of 1 to ensure AA generation. For small molecules, we set the prompt to 0, allowing flexible generation of molecular fragments. We will add a detailed explanation in the revision to clarify this point.

---

### Official Review · Reviewer_tPcN · 2025-03-17

**Overall Recommendation:** 4

**Summary:**

This paper addresses the task of generating de novo binder molecules to target proteins. Importantly, the paper introduces a single unified framework and model, *UniMoMo*, that can generate peptide binders, antibody binders, and small molecule binders. To this end, the paper proposes a variational autoencoder that encodes different types of molecules building block-wise in a unified latent space. Atomistic details are encoded in latent space and reconstructed with an interative decoder. A diffusion model is trained afterwards in latent space for generation, while conditioning on protein target information. Crucially, the paper demonstrates meaningful improvements by training a model jointly on all molecule types, indicating a certain type of knowledge transfer. The authors extensively validate their model and show strong performance, outperforming baselines in many experiments.

**Claims And Evidence:**

Yes, the paper supports all its claim by extensive experiments.

**Essential References Not Discussed:**

I am not aware of any essential but missing references.

**Experimental Designs Or Analyses:**

All experimental designs and analyses seems appropriate to me, as well as sound and valid.

**Methods And Evaluation Criteria:**

Yes, all proposed methods and evaluation criteria make sense to me and are appropriate for the tackled problem. Overall, the method and design of UniMoMo is well motivated, although some of its details seem ad-hoc and could be explained better. See my questions below.

**Other Comments Or Suggestions:**

I believe equation (13) is missing the square root over $\sqrt{1-\bar{\alpha}^t}$, keeping in mind that in equation (12), we have the variance, and when doing reparametrized sampling we need the standard deviation.

**Other Strengths And Weaknesses:**

**Strengths:**
- The paper runs very extensive evaluation experiments for all the different molecule types that UniMoMo can generate. I appreciate the detailed evaluations and the strong results.
- The experiment on the GPCR is very interesting, showing that the model leverages aspects it has learnt from different molecule modalities when generating its small molecule binder.
- The finding that jointly learning a model over different molecule types improves performance in all applications is very interesting and significant, I believe. To the best of my knowledge, this has not been done or shown before in this fashion in a binder generation setting.
- While the overall framework and model is somewhat complex, the paper makes a good job explaining the approach and the supplementary material includes a lot of details. The paper is mostly clear and easy to read.
- The tackled applications, antibody design, small molecule drug design, and peptide design for target proteins are impactful with direct real world applications, which further underscores the relevance of the method.
- UniMoMo builds on established and existing concepts (molecule autoencoders, latent diffusion, etc.), but its detailed architecture, joint building block-wise representation and latent encoding and decoding scheme are novel, to the best of my knowledge.

**Weaknesses:**
- Some method details are not well motivated or explained, see questions below.
- If my understanding is correct, the method only works if the binding site on the target protein is known and given.

While I have several questions about the method and believe that some details could be explained and motivated better, I think UniMoMo is overall a strong model and this is an interesting paper. Hence, I am recommending acceptance.

**Questions For Authors:**

1. In line 096 in the introduction, the authors point out that the method uses E(3)-*invariant* latent states and E(3)-*equivariant* coordinates. This *invariance* and *equivariance* is later in the method section not further discussed. But how exactly is the invariance of the latents guaranteed, as well as the equivariance of the coordinates? I would suggest the authors to discuss this in more detail.
2. My understanding is that UniMoMo tackles the situation where the location of the binding site on the target molecule is known and given. Can the authors clarify? What if we do not know the binding site? Can we still use and apply UniMoMo?
3. The authors construct its latent space through encoding both into "abstract" latents $z_i$ and coordinate-valued latents $\vec{z}_i$. Why exactly is this separation needed? Also, why use coordinate-valued latents $\vec{z}_i$ at all, and why not instead directly use the original building block coordinates, $\vec{X}_i$, together with the other latents $z_i$ to encode all the additional information? I think this could be explained and motivated better, and it also could be interesting to run an ablation study for this modeling choice.
    - Related, an ablation study over the KL weights $\lambda_1$ and $\lambda_2$ in equation (2) would be interesting.
4. The authors manually add extra noise to $\vec{z}_i$ before feeding it to the decoder, to enforce robustness. This seems very ad-hoc and in principle the sampling of the posterior distribution should already lead to noise injection making the decoder robust. A more principled way would be to increase the KL weight, such that the posterior encoding distributions themselves become wider and smoother, as opposed to manually adding noise. I would suggest the authors to better justify their choice here and also show in an ablation study that this is necessary.
5. The authors speak of *"motion vectors"*. If my understanding is correct, this is the vector field that encodes the flow in the flow matching framework. If that is the case, I would suggest the authors to update their wording here, as I have never heard the expression motion vectors for this quantity before.
6. An important detail I am missing: The authors build one joint model for all molecule types, and then apply it to the different applications in the experiments. How exactly is the model told to either generate a peptide, a small molecule, or an antibody for the different applications? Is there some conditioning given to the model to control this? Or does the model generate different molecule types entirely randomly? But that would be confusing, because the model is applied to the specific applications. I think I am missing something here and I would suggest the authors to be clearer about that.

**Relation To Broader Scientific Literature:**

The paper is appropriately positioned with respect to the broader literature. Most importantly, the paper discusses many previous related works tackling structure-based drug design.

While there already exist related frameworks, including latent diffusion models, for similar molecular modeling tasks, the particular framework proposed by the authors seems novel. The way the variational autoencoder is designed in this paper and the unified decomposition of different types of molecules into graphs of coarse building blocks seem new to me.

**Theoretical Claims:**

The paper does not rely on complex proofs, novel mathematical frameworks, or theoretical claims. Rather, the paper proposes a novel unified molecule generation system.

---

> ### Author Rebuttal · Authors · 2025-03-31
>
> Thanks for your appreciation and insightful feedback, which is very helpful in improving our paper!
> > W & Q2: If my understanding is correct, the method only works if the binding site on the target protein is known. What if we do not know the binding site?
>
> Yes, your understanding is correct. Our method requires prior knowledge of the binding site, in line with the convention established in previous studies [a], since most public benchmarks are built upon this setting. This is also biologically reasonable. For example, to study a protein-protein interaction, a binder is often designed at the interface to inhibit the PPI. If truely no biological knowledge is available, pocket detection tools like MaSIF [b] should be able to identify potential sites for pocket-based design models.
>
> [a] Pocket2mol: Efficient molecular sampling based on 3d protein pockets. ICML 2022
>
> [b] Deciphering interaction fingerprints from protein molecular surfaces using geometric deep learning. Nature Methods
> > C: Equation (13) is missing the square root over $1-\bar{\alpha}^t$.
>
> Thanks for catching this! We apologize for the typo and will correct it in the revision
> > Q1: How exactly is the invariance of the latents guaranteed, as well as the equivariance of the coordinates?
>
> For the diffusion part, GeoDiff [c] proves that the diffusion process maintain equivariance via an equivariant denoising kernel (Proposition 1), which in our work is the equivariant transformer[d]. The transformer is scalar-based[e], which maintain the equivariance with inner product and unbiased linear transformation on the velocities.
>
> For the VAE part, the encoding equivariance is determined by the network used, which is also an equivariant transformer. The decoder employs a short flow matching, with a similar theoretical foundation as GeoDiff on equivariant flow matching [f] (Theorem 4.1).
>
> Thanks for pointing this out! We will add a detailed discussion in the revision for clarity.
>
> [c] GeoDiff: A Geometric Diffusion Model for Molecular Conformation Generation. ICLR 2022.
>
> [d] An Equivariant Pretrained Transformer for Unified 3D Molecular Representation Learning. preprint.
>
> [e] Scalars are universal: Equivariant machine learning, structured like classical physics. NeurIPS 2021.
>
> [f] Equivariant Flow Matching with Hybrid Probability Transport for 3D Molecule Generation. NeurIPS 2023.
> > Q3: Why not use the original building block coordinates $\vec{X}_i$, together with the other latents $z_i$ to encode all the additional information? Related, an ablation study over the KL weights $\lambda_1$ and $\lambda_2$ in equation (2) would be interesting.
>
> Thanks for the valuable question. The block coordinates $\vec{X}_i\in\mathbb{R}^{n_i\times 3}$ vary in length due to different number of atoms per block (e.g. residue), which makes direct diffusion challenging, as it typically operates in a fixed-length space. A similar issue arises for $H_i$. Thus, we use an all-atom VAE to compress these irregular matrices into fixed-length latent vectors $z_i$ and $\vec{z}_i$, making diffusion feasible.
>
> Regarding $\lambda_1$ and $\lambda_2$, this is a really insightful question. A higher $\lambda$ smooths the latent space, improving continuity for later generative modeling but also increasing compression, which may limit expressivity. Thus, the weight can neither be too low nor be too high, as shown in the [validation loss curves](https://anonymous.4open.science/r/UniMoMo-CEA0/assets/l1l2.png). Ultimately, we select the combination with the lowest validation loss.
> > Q4: The authors manually add extra noise to $\vec{z}_i$ before feeding it to the decoder, to enforce robustness. A more principled way would be to increase the KL weight.
>
> Thanks for the suggestion. The extra noises primarily benefits small molecules, which have more intricate inter-block connections than peptides and antibodies connected by peptide bonds, thus require higher robustness to coordinate errors introduced by diffusion. Given that the weight of KL loss is already high (0.8), further increasing it would overly constrain the latent space (as in Q3 above and the table below).
> |model|vina(score only)|vina(minimize)|vina(dock)|
> |-|-|-|-|
> |w/o extra noise|-4.62|-5.58|-7.24|
> |w/o extra noise+KL 1.0|-4.42|-5.59|-7.09|
> |w/ extra noise|-5.72|-6.08|-7.25|
> > Q5: Are "motion vectors" the vector field in the flow matching framework?
>
> Yes, we will replace them with "vector fields" in the revision to avoid ambiguity.
> > Q6：How exactly is the model told to either generate a peptide, a small molecule, or an antibody for the different applications?
>
> Thanks for the insightful question! We apologize for not making this clear in the paper. We assign a binary prompt to each block, with 1 indicating amino acid (AA) and 0 without restriction. For peptide and antibody benchmarks, all blocks are assigned 1 to ensure AA generation. For small molecules, we use 0 to allow arbitrary fragment generation. We will clarify this in the revision.

---

### Decision · Program_Chairs · 2025-05-01

**Decision:**

Accept (poster)

**Comment:**

This paper received reasonably positive reviews. The reviewers are not very enthusiastic, mainly due to the marginal improvement in certain cases, but have expressed consistent support. Thus an accept is recommended.